# Translational control in the spinal cord regulates gene expression and pain hypersensitivity in the chronic phase of neuropathic pain

Kevin C Lister[1], Calvin Wong[1], Sonali Uttam[1], Marc Parisien[1,2,3], Patricia Stecum[1], Nicole Brown[1], Weihua Cai[1], David Ho-Tieng[1], Mehdi Hooshmandi[1], Ning Gu[1], Mehdi Amiri[4], Francis Beaudry[5,6], Seyed Mehdi Jafarnejad[7], Diana Tavares-Ferreira[8], Nikhil Nageshwar Inturi[8], Khadijah Mazhar[8], Hien T Zhao[9], Bethany Fitzsimmons[9], Christos G Gkogkas[10], Nahum Sonenberg[4], Theodore J Price[8], Luda Diatchenko[1,2,3], Yaser Atlasi[7], Jeffrey S Mogil[1,3,11], Arkady Khoutorsky[1,2,3]*

[1]Department of Anesthesia, McGill University, Montreal, Canada; [2]Faculty of Dental Medicine and Oral Health Sciences, McGill University, Montreal, Canada; [3]Alan Edwards Centre for Research on Pain, McGill University, Montreal, Canada; [4]Department of Biochemistry and Goodman Cancer Research Centre, McGill University, Montreal, Canada; [5]Département de biomédecine vétérinaire, Faculté de médecine vétérinaire, Université de Montréal, Montreal, Canada; [6]Centre de recherche sur le cerveau et l'apprentissage (CIRCA), Université de Montréal, Montréal, Canada; [7]Patrick G. Johnston Centre for Cancer Research, Queen's University Belfast, Belfast, United Kingdom; [8]Department of Neuroscience and Center for Advanced Pain Studies, University of Texas at Dallas, Dallas, United States; [9]Ionis Pharmaceuticals, Inc, Carlsbad, United States; [10]Biomedical Research Institute, Foundation for Research and Technology-Hellas, University Campus, Ioannina, Greece; [11]Department of Psychology, Faculty of Science, McGill University, Montreal, Canada

*For correspondence:
arkady.khoutorsky@mcgill.ca

## eLife Assessment

Using a combination of innovative and robust techniques, this study outlines cell-type-specific translational landscape changes that occur in the spinal cord neurons in the early and late phases of nerve injury. The authors provided **compelling** evidence suggesting an essential role of protein synthesis regulation in the chronic phase of neuropathic pain. Although additional mechanisms contributing to late-phase neuropathic pain beyond altered PV+ neuron excitability remain to be elucidated, this is a **fundamental** and significant study toward a comprehensive understanding of the molecular pathways involved in neuropathic pain.

**Abstract** Sensitization of spinal nociceptive circuits plays a crucial role in neuropathic pain. This sensitization depends on new gene expression that is primarily regulated via transcriptional and translational control mechanisms. The relative roles of these mechanisms in regulating gene expression in the clinically relevant chronic phase of neuropathic pain are not well understood. Here, we show that, in mice, changes in gene expression in the spinal cord during the chronic phase of

neuropathic pain are substantially regulated at the translational level. Downregulating spinal translation at the chronic phase alleviated pain hypersensitivity. Cell type-specific profiling revealed that spinal inhibitory and excitatory neurons exhibited substantial changes in translation after peripheral nerve injury. Notably, increasing translation selectively in all inhibitory neurons or parvalbumin-positive (PV⁺) interneurons, but not excitatory neurons, promoted mechanical pain hypersensitivity. Furthermore, increasing translation in PV⁺ neurons decreased their intrinsic excitability and spiking activity. Conversely, reducing translation in spinal PV⁺ neurons prevented the nerve injury-induced decrease in excitability but did not alleviate mechanical hypersensitivity. Together, these findings advance our understanding of translational control mechanisms in the spinal cord during neuropathic pain and highlight their cell type- and phase-specific contributions to gene expression and pain hypersensitivity.

## Introduction

Peripheral nerve injury may result in neuropathic pain, a debilitating condition with limited effective treatment options (*Finnerup et al., 2021*; *Colloca et al., 2017*; *Costigan et al., 2009*). The development (early) and maintenance (late) phases of neuropathic pain are mediated by structural and functional changes in peripheral and central pain-processing compartments via complex interactions between neuronal and non-neuronal cells (*Costigan et al., 2009*; *Chen et al., 2018*; *Peirs and Seal, 2016*; *Peirs et al., 2021*). The persistence of these changes relies on de novo gene expression, which is tightly regulated, primarily via transcriptional and translational control mechanisms. Whereas previous studies have characterized transcriptional (*LaCroix-Fralish et al., 2011*; *Ray et al., 2023*; *North et al., 2019*; *Ghazisaeidi et al., 2023*; *Barry et al., 2023*) and translational (*Uttam et al., 2018*; *Megat et al., 2019*) changes in the dorsal root ganglia (DRG) and spinal cord following peripheral nerve injury, demonstrating their important roles during the early stage of neuropathic pain (*Uttam et al., 2018*; *Géranton et al., 2009*), the investigations of these mechanisms in the late maintenance phase are lacking.

Studies in neuronal and non-neuronal cells have revealed a poor correlation between the expression levels of distinct mRNAs and the abundance of their corresponding proteins (*Taniguchi et al., 2010*; *Schwanhäusser et al., 2011*). The regulation of mRNA translation significantly affects the cellular proteome, representing an important mechanism to account for the discordance between mRNA and protein expression (*Bourke et al., 2023*; *Khoutorsky and Price, 2018*). Thus, it is essential to investigate the role of translational control in regulating gene expression and pain hypersensitivity during the clinically relevant late phase of neuropathic pain.

Recent methodological advances enable the investigation of genome-wide transcriptional and translational changes (using ribosome profiling [Ribo-seq]; *Ingolia et al., 2012*), as well as the identification of actively translating mRNAs in specific cell types (using translating ribosome affinity purification [TRAP]; *Heiman et al., 2014*). Here, we employed Ribo-seq and TRAP techniques to study alterations in gene expression in the DRG and spinal cord during early and late phases of neuropathic pain. We found that both transcriptional and translational mechanisms regulate changes in gene expression in the DRG in the early phase (4 days after nerve injury) and the late phase (63 days after nerve injury), as well as in the spinal cord in the early phase. Surprisingly, changes in gene expression in the spinal cord in the late phase of neuropathic pain were regulated more extensively at the translational level. Targeting a key translation initiation factor, eukaryotic translation initiation factor 4E (eIF4E), in the spinal cord provided a long-lasting alleviation of evoked and spontaneous pain during the maintenance phase. Cell type-specific translational profiling, using metabolic labeling and TRAP, revealed greater nerve injury-induced translational changes in spinal inhibitory neurons than in excitatory neurons. Activating translation in all inhibitory or in parvalbumin-positive (PV⁺) interneurons, but not in excitatory neurons, was sufficient to induce mechanical hypersensitivity. However, while inhibiting translation in PV neurons prevented the nerve injury-induced decrease in PV neuron excitability, it was not sufficient to alleviate mechanical hypersensitivity.

Taken together, this study provides a characterization of translational changes in the early and chronic phases of neuropathic pain and reveals a role for spinal translational control in the maintenance of pain hypersensitivity.

## Results

### Translational regulation of gene expression in the chronic phase of neuropathic pain

To study changes in gene expression at both transcriptional (transcriptome) and translational (translatome) levels, we employed Ribo-seq on DRG and lumbar spinal cord tissue obtained from mice subjected to an experimental assay of peripheral nerve damage-induced (i.e., neuropathic) pain, spared nerve injury (SNI, *Figure 1A*; *Decosterd and Woolf, 2000*), or sham surgery. SNI prominently features mechanical hypersensitivity, which develops within 2–4 days of the nerve injury and persists for many months (*Millecamps et al., 2023*). L3–L5 DRGs and the ipsilateral dorsal half of the corresponding segment of the lumbar spinal cord (illustrated in a schematic diagram in *Figure 1A*) were collected on day 4 (early phase) and day 63 (late phase) post-SNI and processed for Ribo-seq. Ribo-seq allows the identification of mRNA fragments (ribosome footprints [rFPs]) where translating ribosomes are bound. These mRNA fragments thus remain protected from nuclease-mediated RNase degradation, and thereby reveal the number and location of ribosomes on specific transcripts (*Figure 1B*; *Ingolia et al., 2012*). Normalization to the corresponding transcript abundance from the parallel mRNA-seq analysis provides a measure of mRNA translation efficiency on a genome-wide scale. Using this approach, we identified transcriptionally and translationally regulated genes in each tissue and time point ($n$ = 3 biological replicates/condition, 15 mice pooled per replicate). In the DRG, a significant number of transcripts were altered at both transcriptional and translational levels at day 4 post-SNI (*Figure 1C, D*; *Figure 1—figure supplement 1A* provides volcano plots for all conditions; full datasets are provided in *Supplementary file 1*). At day 63 post-SNI, the number of transcriptionally altered mRNAs in the DRG decreased compared to day 4 (*Figure 1C, D*; *Figure 1—figure supplement 1B*). In the spinal cord, changes in gene expression were less pronounced and on day 4 post-SNI, a comparable number of differentially transcribed genes and differentially translated genes were identified (*Figure 1D, E*; *Figure 1—figure supplement 1A*). Surprisingly, on day 63 post-SNI, changes in gene expression in the spinal cord occurred predominantly at the translational but not transcriptional level (*Figure 1D, E*; *Figure 1—figure supplement 1B*). Gene Ontology pathway analysis of translationally regulated genes in the spinal cord on day 63 post-SNI revealed alterations in processes related to extracellular matrix organization and its interaction with cell surface receptors, cell adhesion, and protein turnover (*Figure 1F*). Collectively, these results indicate that both transcriptional and translational mechanisms mediate changes in gene expression in the DRG in both the early and late phases of neuropathic pain and in the spinal cord in the early phase. In the late chronic phase, however, changes in gene expression in the spinal cord are controlled more prominently at the translational level.

### Suppression of spinal translation alleviates established pain hypersensitivity

The important role of spinal translation in regulating changes in gene expression in the chronic phase of neuropathic pain prompted us to test whether targeting translation in the spinal cord can alleviate established pain hypersensitivity. Translation initiation and the activity of cap-binding protein eIF4E, which facilitates the recruitment of ribosomes to the mRNA, are rate-limiting steps in protein synthesis (*Gingras et al., 2001*; *Tahmasebi et al., 2018*). eIF4E is a key translation initiation factor regulating pain-related plasticity as it integrates information from two signaling pathways that are activated in neuropathic pain to stimulate mRNA translation and promote pain hypersensitivity: The mechanistic target of rapamycin complex 1 (mTORC1), and mitogen-activated protein kinases (MAPKs), such as ERK and p38 (*Khoutorsky and Price, 2018*; *Ji et al., 2009*; *Melemedjian and Khoutorsky, 2015*; *Figure 2A*). Moreover, previous studies have revealed that a partial reduction in eIF4E expression (e.g., in *Eif4e*[+/−] mice and in mice treated with eIF4E shRNA) is both well-tolerated and alleviates adverse phenotypes in cancer (*Truitt et al., 2015*) and autism spectrum disorder (*Gkogkas et al., 2013*; *Santini et al., 2013*) mouse models. To study the role of mRNA translation in chronic pain hypersensitivity, we used an antisense oligonucleotide (ASO) against mouse *Eif4e* to modulate eIF4E expression. To target eIF4E in the central nervous system but not the DRG, we injected eIF4E-ASO via the intracerebroventricular (i.c.v., 100 mg/kg) route (*Mohan et al., 2018*), resulting in a ~38% reduction in eIF4E protein expression in the lumbar spinal cord 2 weeks post-injection (*Figure 2B*), without changing eIF4E levels in the DRG (*Figure 2C*). We first assessed the effect of eIF4E downregulation

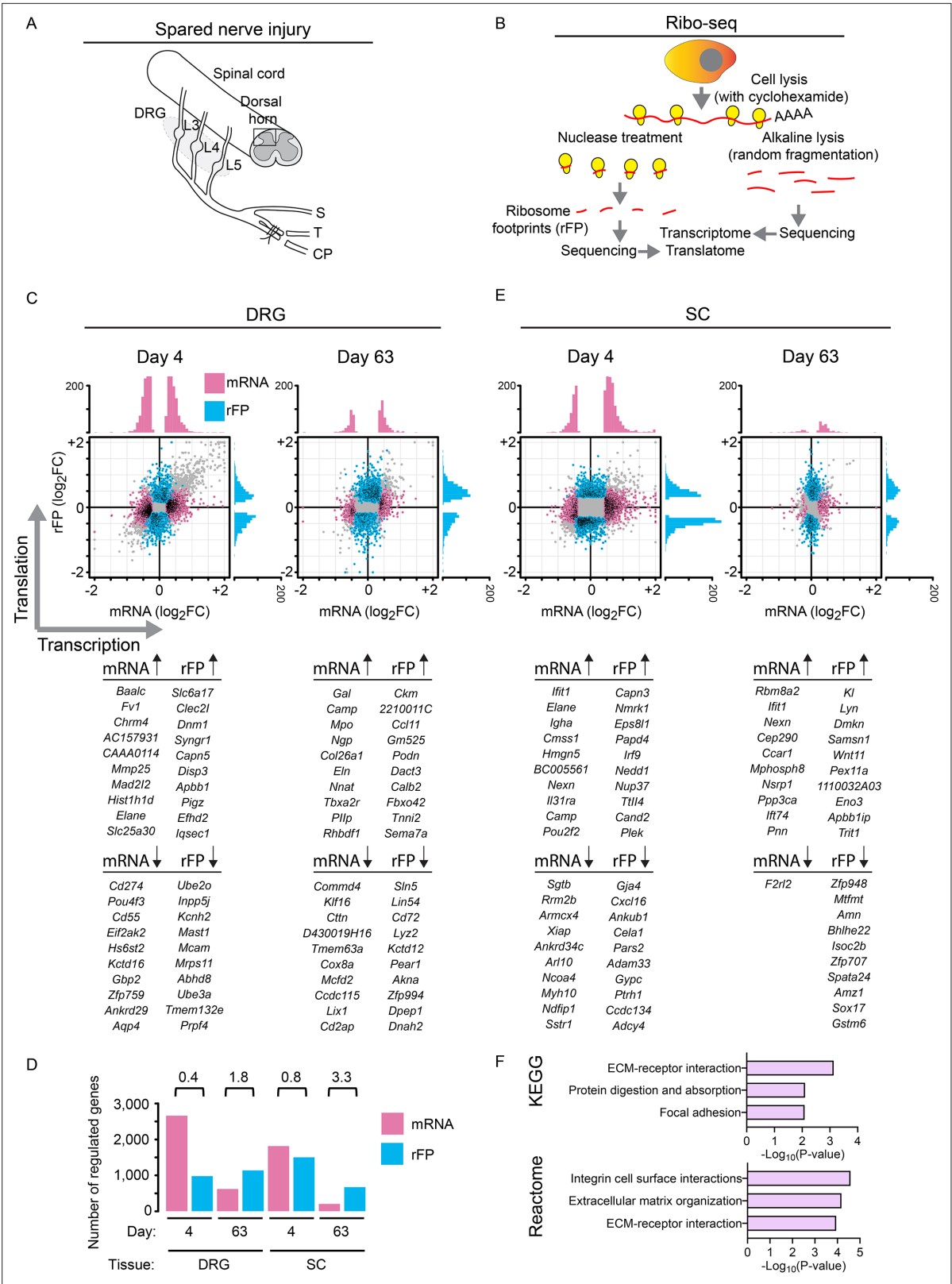

**Figure 1.** Transcriptional and translational analysis of gene expression using Ribo-seq. (**A**) A schematic diagram showing the spared nerve injury (SNI) assay. S: sural branch, T: tibial branch, and CP: common peroneal branch. (**B**) An illustration of the ribosome profiling technique (Ribo-seq). Scatter plot shows ribosomal footprint (rFP) log$_2$ fold change (FC), reflecting translational changes, as a function of mRNA log$_2$ fold change for dorsal root ganglia (DRG, **C**) and spinal cord (SC, **E**), at day 4 and 63 post-SNI in female mice. Each dot is a gene. Fold change evaluated between SNI and sham conditions.

*Figure 1 continued on next page*

*Figure 1 continued*

Color coding indicates modality of differential gene expression control, either at the transcriptional level (mRNA, magenta) or at the translational level (rFP, blue). Under each scatter plot, a list of the top 10 upregulated and downregulated genes (at the mRNA and rFP levels) is shown for each condition. *Figure 1—figure supplement 1* shows volcano plots for all conditions. *Supplementary file 1* includes complete datasets (worksheets 1–4), with yellow highlighting category of change in gene expression (translation only, transcription only, stable, opposite change, and homodirectional), gray indicating mRNA log$_2$FC, and blue indicating rFP log$_2$FC. (**D**) Number of genes showing changes at mRNA and rFP levels across independent biological replicates. The rFP/mRNA ratio for each condition is shown above the columns. (**F**) Pathway analyses of translationally regulated genes in the SC at day 63 post-SNI in the Kyoto Encyclopedia of Genes and Genomes (KEGG) and Reactome databases.

The online version of this article includes the following figure supplement(s) for figure 1:

**Figure supplement 1.** Volcano plots showing changes in mRNA (top), ribosome footprint (rFP, middle), and translational efficiency (TE, bottom) levels in the dorsal root ganglia (DRG) and SC tissues at day 4 post-spared nerve injury (SNI) (**A**) and day 63 post-SNI (**B**).

on established pain hypersensitivity. Injection of eIF4E-ASO at week 6 post-SNI alleviated mechanical pain hypersensitivity in the von Frey test 2 weeks later (at week 8 post-SNI; the experimental time course is provided in *Figure 2D*, von Frey data in *Figure 2E*). Reduced hypersensitivity persisted for 4 additional weeks (up to week 12 post-SNI), demonstrating a long-lasting effect following a single eIF4E-ASO administration. eIF4E-ASO also attenuated spontaneous pain, as assessed using the Mouse Grimace Scale (MGS) on week 8 post-SNI (*Figure 2F*).

We then assessed the effect of downregulating eIF4E during the early development phase of neuropathic pain by administering eIF4E-ASO and control-ASO 10 days before the peripheral nerve injury. Surprisingly, we found no alleviation of mechanical hypersensitivity at day 4 post-SNI in eIF4E-ASO-injected mice (the experimental time course is provided in *Figure 2G*, von Frey data in *Figure 2H*). However, testing at later time points showed that mice injected with eIF4E-ASO exhibited reduced mechanical pain hypersensitivity at week 2 after the nerve injury (*Figure 2H*), and the effect became more pronounced at weeks 4 and 8. The MGS was also reduced in eIF4E-ASO-injected mice at week 4 and 8 post-SNI (*Figure 2I*). To control for potential non-specific effects of eIF4E-ASO following i.c.v. administration, we performed rotarod and open field tests, which revealed no differences in locomotor function between mice injected with eIF4E-ASO and control-ASO (*Figure 2J, K*). Together, these results indicate that downregulation of eIF4E in the spinal cord using ASO alleviates pain hypersensitivity in the late but not the acute stages of neuropathic pain.

## Cell type-specific translational profiling after peripheral nerve injury

Ribo-seq provided a comprehensive characterization of translational landscape in DRG and spinal cord tissues during the early and late phase of neuropathic pain. However, this approach does not allow the measurement of gene expression in distinct neuronal subtypes and cannot distinguish between neuronal and non-neuronal cells. To assess protein synthesis in specific neuronal subpopulations, we used fluorescence noncanonical amino acid tagging (FUNCAT) (*Hooshmandi et al., 2024*; *Dieterich et al., 2010*), focusing on two major neuronal subtypes in the spinal cord, excitatory and inhibitory neurons. In FUNCAT, mice are injected with a noncanonical amino acid, azidohomoalanine (AHA), which is charged onto methionine tRNA and incorporated into newly synthesized proteins (*Figure 3A*). Visualization of AHA incorporation using click chemistry and fluorescent labeling provides a measure of de novo general protein synthesis in spinal cord sections. The specificity of this approach was validated using the protein synthesis inhibitor anisomycin, which blocked AHA incorporation in the spinal cord (*Figure 3B*). FUNCAT analysis showed that AHA incorporation increased at day 4 and 60 post-SNI in Pax2[+] inhibitory neurons (day 4: *Figure 3C*; day 60: *Figure 3D*), whereas no statistically significant changes were found in excitatory neurons (NeuN[+] and Pax2[−]).

Next, we employed the TRAP approach to identify specific mRNAs that are actively translated in excitatory and inhibitory neurons. In TRAP, the eGFP-tagged ribosomal protein, L10a, is expressed in a genetically defined cellular population (via a specific gene promoter), followed by IP of tagged ribosomes with an anti-eGFP antibody and the sequencing of ribosome-bound mRNAs (*Figure 4A*). We performed TRAP analysis on two major subpopulations of neurons: a subset of excitatory neurons, defined by Tac1 (using *L10a*-eGFP; *Tac1*[Cre] mice); and inhibitory neurons, defined by GAD2 (using *L10a*-eGFP; *Gad2*[Cre] mice). Tac1 is expressed in a subset of excitatory interneurons and projection neurons in the spinal cord that play important roles in processing nociceptive information, as well as driving spinal plasticity and chronic pain-related behaviors (*Barik et al., 2021*; *Huang et al., 2019*).

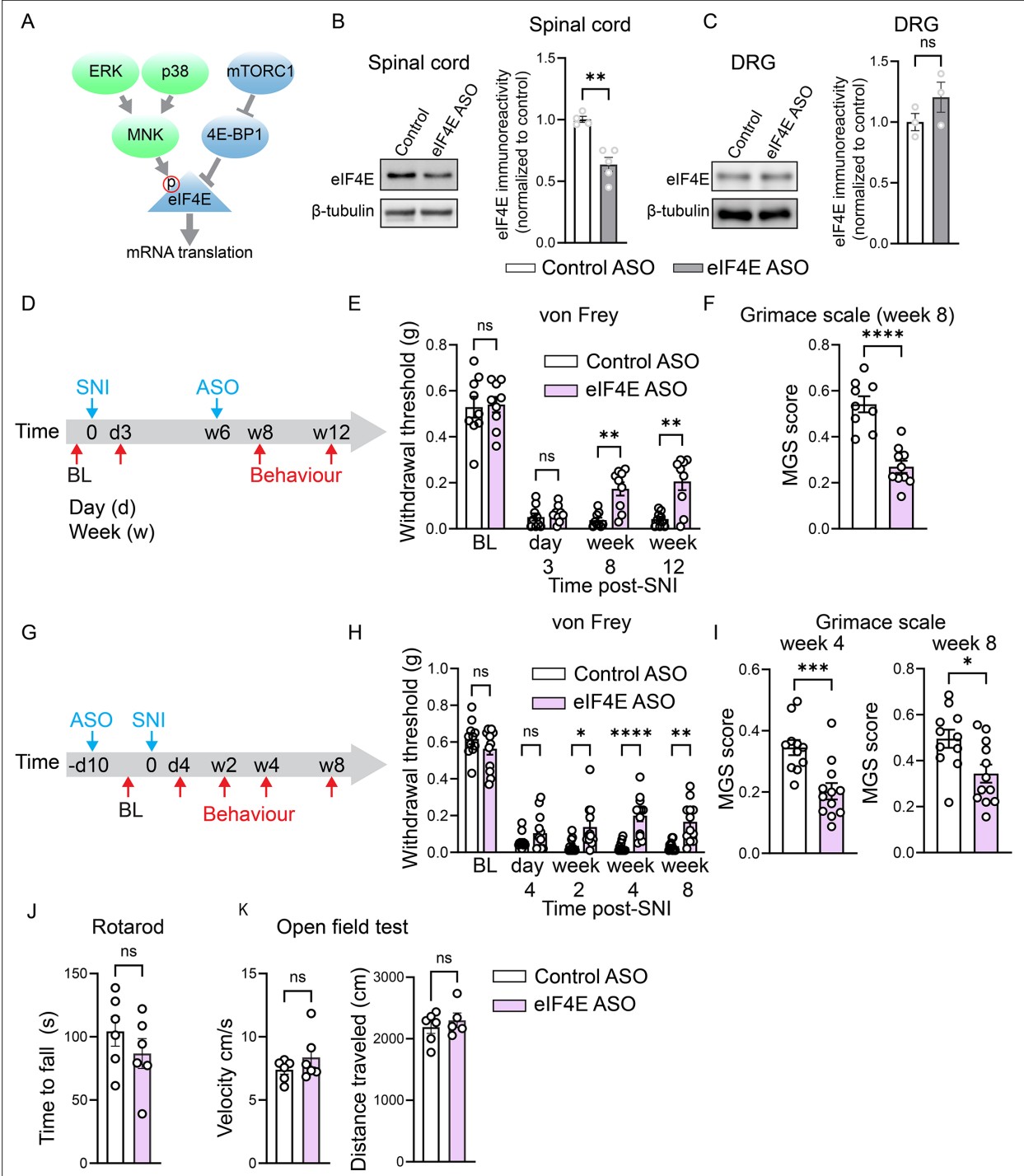

**Figure 2.** Targeting spinal translation alleviates pain hypersensitivity at the late stage after peripheral nerve injury. (**A**) A schematic showing the regulation of eIF4E via mTORC1/4E-BP1 and MAPKs/MNK pathways. eIF4E-ASO (i.c.v.) reduces eIF4E protein levels in the spinal cord (**B**) but not dorsal root ganglia (DRG) (**C**) 2 weeks after administration (n = 3–4/group). (**D**) Time course of ASO (eIF4E and control) administration after spared nerve injury (SNI). The effect of ASO on von Frey (50% withdrawal threshold: **E**, n = 9/group) and Mouse Grimace Scale (MGS) (**F**, n = 9/10 mice per group). (**G**) Time course of ASO administration before SNI and its effect on the von Frey (50% withdrawal threshold: **H**, n = 11/12 mice per group) and MGS (**I**, n = 11/12 mice per group) tests. An unpaired two-tailed *t*-test was used in B, C, F, and I. Two-way ANOVA followed by Tukey's post hoc comparison was used in E and H. Each data point represents an individual animal. A comparable number of male and female mice was used in all experiments. Data are plotted as mean ± SEM. *p < 0.05, **p < 0.01, ***p < 0.001, ****p < 0.0001, ns – not significant.

The online version of this article includes the following source data for figure 2:

*Figure 2 continued on next page*

*Figure 2 continued*

**Source data 1.** PDF file containing original western blots for *Figure 2B, C*, indicating the relevant bands and treatments.

**Source data 2.** Original files for western blot analysis displayed in *Figure 2B, C*.

GAD2⁺ neurons encompass numerous subpopulations of spinal cord inhibitory neurons that are critical for the development and maintenance of neuropathic pain (*Todd, 2010*; *Peirs et al., 2020*). To this end, *L10a*-eGFP; *Tac1*^Cre^ and *L10a*-eGFP; *Gad2*^Cre^ mice were subjected to SNI or sham surgery (bilaterally), and lumbar dorsal spinal cord tissue was collected at days 4 and 60 after the nerve injury. mRNAs isolated from the immunoprecipitated (IP) and input (IN) samples were sequenced. Expression levels (IP/IN) of excitatory (*Slc17a7*, *Tac1*, *Cck*, *Nts*) and inhibitory (*Slc32a1*, *Gad2*, *Pax2*, *Pvalb*) neuronal markers, as well as markers of non-neuronal cells (e.g., *Aldh1l1*, *Gfap*, *Tmem119*, *Cx3cr1*), are shown in *Figure 4B* for *L10a*-eGFP; *Gad2*^Cre^ mice and in *Figure 4C* for *L10a*-eGFP; *Tac1*^Cre^ mice, demonstrating the specificity of the approach. Changes in ribosome occupancy were found in 126 mRNAs in the early phase and 223 mRNAs in the late phase in GAD2⁺ neurons, and 118 mRNAs in the early phase and 161 in the late phase in Tac1⁺ neurons (IP: *Figure 4D–H*, IP/IN datasets are provided

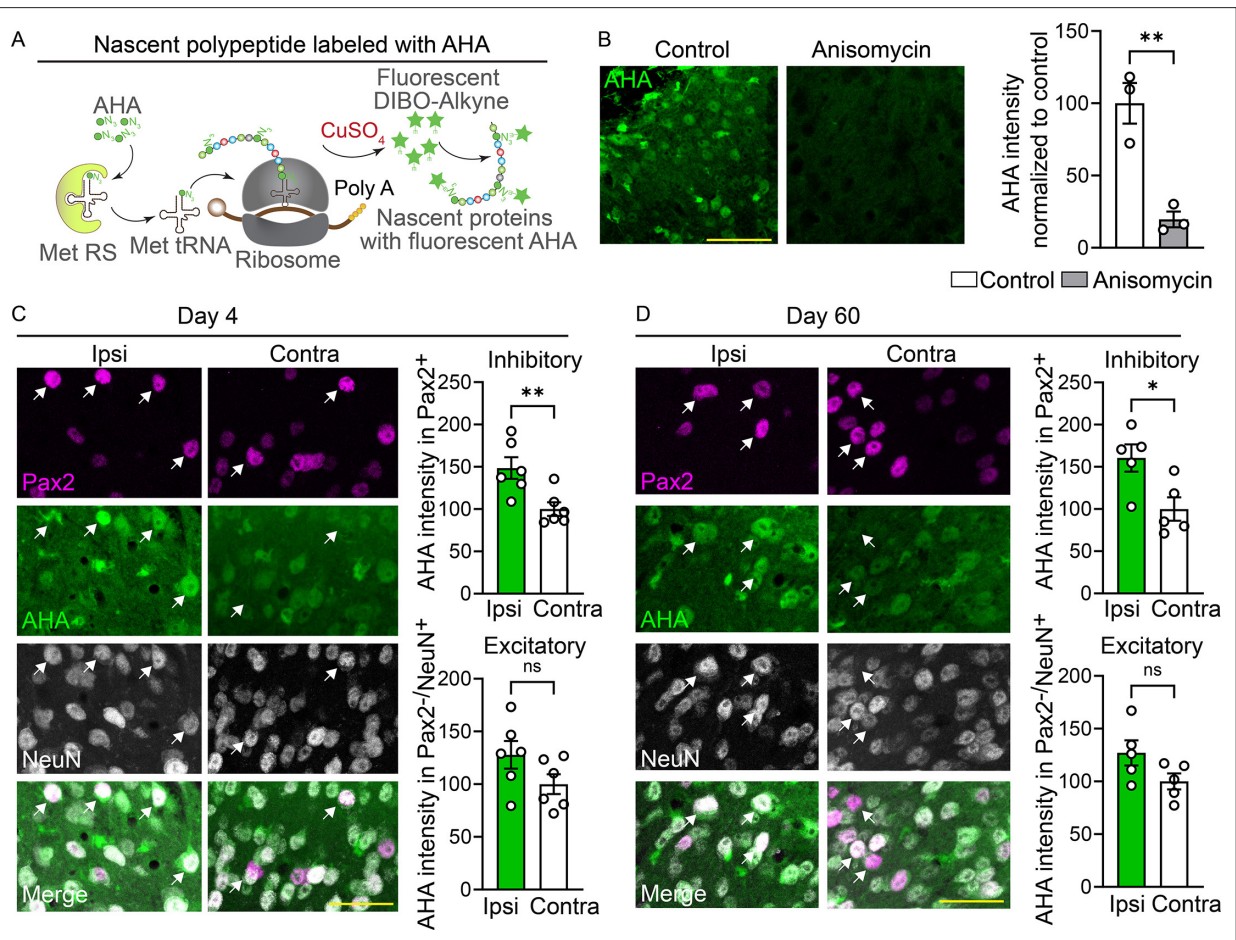

**Figure 3.** Assessment of protein synthesis using metabolic labeling. (**A**) Illustration of protein synthesis assessment using FUNCAT. (**B**) Anisomycin (100 mg/kg, i.p. injection 1 hr before azidohomoalanine [AHA] injection) treatment blocked AHA incorporation (*n* = 3 female mice per group, normalized to the control group), demonstrating the validity of the approach. AHA signal in the superficial spinal cord (laminae I–III, defined based on NeuN staining) was quantified in inhibitory neurons (Pax2⁺, examples marked by white arrow) and excitatory neurons (Pax2⁻/NeuN⁺) at day 4 (**C**, *n* = 6 female mice per group) and day 60 (**D**, *n* = 5 female mice per group) post-spared nerve injury (SNI). AHA signal intensity (integrated density on maximum-intensity projection images) in the soma of inhibitory and excitatory neurons was averaged across 25 cells/mouse to obtain a single value for each mouse (see Methods for details of the analysis). Ipsi indicates ipsilateral and Contra indicates contralateral to the site of injury. Scale bars: 50 µm for B and 30 µm for C, D. An unpaired two-tailed *t*-test was used. Each data point represents an individual animal. Data are plotted as mean ± SEM. *p < 0.05, **p < 0.01, ns – not significant.

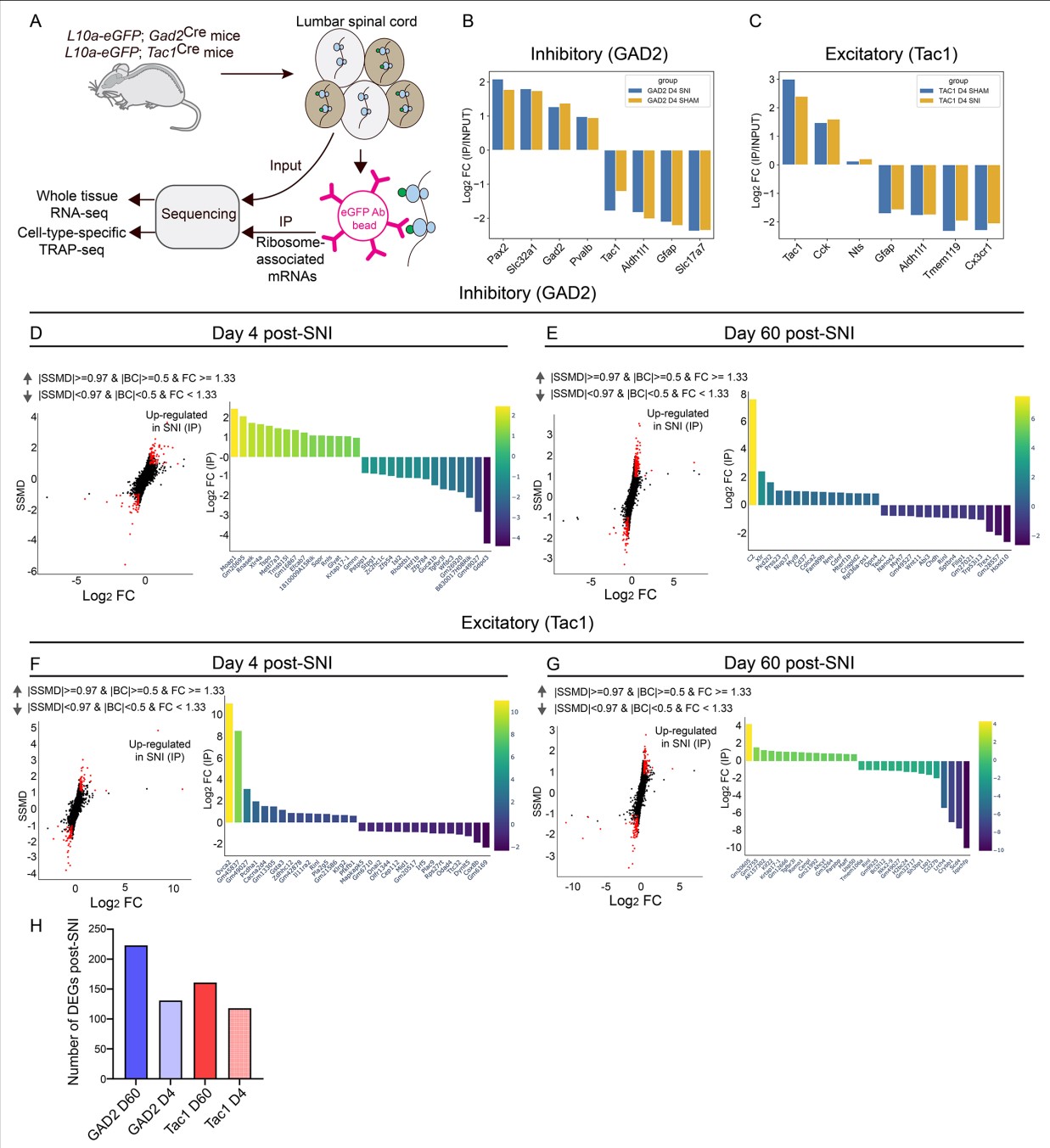

**Figure 4.** Cell type-specific profiling of spinal gene expression after peripheral nerve injury. (**A**) A schematic illustrating the TRAP approach to assess gene expression in specific cell types. Confirmation of the specificity of IP fractions for inhibitory neurons in the *L10a*-eGFP; *Gad2*Cre mouse line (**B**) and for excitatory neurons in the *L10a*-eGFP; *Tac1*Cre mouse line (**C**). Experiments were performed in female mice. Dual flashlight plots (left) show the strictly standardized mean difference (SSMD) versus $\log_2$ FC for genes in IP samples and panels on the right show the top 15 upregulated and downregulated genes for inhibitory neurons at day 4 (**D**) and 60 (**E**), and Tac1+ excitatory neurons at day 4 (**F**) and 60 (**G**) post-spared nerve injury (SNI). Positive $\log_2$ FC indicates increased expression in SNI compared to sham mice. Parameters for defining data as upregulated in SNI are indicated at the top. *Supplementary file 1* includes complete datasets (worksheets 5–8), with yellow highlighting $\log_2$ FC values in IP samples and orange highlighting $\log_2$ FC values in IN samples. (**H**) The number of altered genes in each condition (GAD2 D60: SNI versus sham day 60 in GAD2+ neurons; GAD2 D4: SNI versus sham day 4 in GAD2+ neurons; Tac1 D60: SNI versus sham day 60 in Tac1+ neurons; and Tac1 D4: SNI versus sham day 4 in Tac1+ neurons).

in *Supplementary file 1*), suggesting higher translational changes at day 60 post-SNI in GAD2$^+$ inhibitory neurons compared to Tac1$^+$ excitatory neurons. Together, these results establish translational changes in GAD2$^+$ and Tac1$^+$ neurons in the early and late phases of neuropathic pain.

## Upregulating eIF4E-dependent translation in inhibitory neurons promotes pain hypersensitivity

The pronounced upregulation of mRNA translation in inhibitory and excitatory neurons after peripheral nerve injury prompted us to study its functional role in mediating pain hypersensitivity. mTORC1, a master regulator of mRNA translation, stimulates protein synthesis via phosphorylation of the translational repressor eIF4E-binding proteins (4E-BPs), triggering their dissociation from eIF4E to allow cap-dependent translation initiation (*Figure 5A*). Accordingly, deletion of 4E-BPs, which mimics the activation of the mTORC1–eIF4E axis, stimulates translation. There are three isoforms of 4E-BP (4E-BP1, 4E-BP2, and 4E-BP3), which exhibit similar functions but have different tissue distribution (*Khoutorsky et al., 2015*). 4E-BP1 is the main isoform in the pain pathway, as 4E-BP1, but not 4E-BP2 whole-body knockout mice show mechanical pain hypersensitivity (*Khoutorsky et al., 2015*), while 4E-BP3 expression is very low in the nervous system. To increase translation selectively in inhibitory or excitatory neurons, we generated mice lacking 4E-BP1 in each cell type (confirmation of 4E-BP1 downregulation is shown in *Figure 5—figure supplement 1A–D*). Deletion of 4E-BP1 in inhibitory neurons (*Eif4ebp1*$^{fl/fl}$;*Gad2*$^{Cre}$) induced mechanical hypersensitivity without affecting heat sensitivity (*Figure 5B*). A subpopulation of inhibitory neurons, PV$^+$ interneurons, specifically gate mechanical allodynia (*Petitjean et al., 2015*; *Boyle et al., 2019*; *Cao et al., 2022*). Peripheral nerve injury induces substantial plasticity in PV neurons, resulting in a decrease in their intrinsic excitability and spiking activity, and the consequent disinhibition of postsynaptic PKCγ interneurons and engagement of myelinated primary afferents in spinal nociceptive circuits (*Petitjean et al., 2015*; *Boyle et al., 2019*; *Cao et al., 2022*). Deletion of 4E-BP1 in PV neurons (*Eif4ebp1*$^{fl/fl}$;*Pvalb*$^{Cre}$) induced robust mechanical hypersensitivity, similar to that observed in *Eif4ebp1*$^{fl/fl}$;*Gad2*$^{Cre}$ mice (*Figure 5C*; no change was found in heat sensitivity). Recording from lumbar spinal cord slices showed that PV neurons lacking 4E-BP1 exhibit reduced excitability, as evident by a decreased firing rate in response to a depolarization pulse (*Figure 5D*) and elevated rheobase compared to PV neurons from control mice (*Figure 5E*). No change was observed in membrane capacitance (*Figure 5F*), resting membrane potential (*Figure 5G*), and input resistance (*Figure 5H*). To study the role of translation in peripheral nerve injury-induced plasticity in spinal PV neurons, we selectively downregulated eIF4E in PV neurons in the lumbar spinal cord before SNI. To this end, an adeno-associated virus (AAV)-expressing shRNAmir against eIF4E (AAV9-CAG-DIO-eGFP-eIF4E-shRNAmir) was injected into the lumbar dorsal horn parenchyma of *Pvalb*$^{Cre}$ mice 14 days before the SNI (*Figure 5I* shows experimental design, *Figure 5—figure supplement 1E* shows confirmation of reduced eIF4E levels). Recordings from PV neurons in spinal cord slices revealed that downregulation of eIF4E in PV neurons prevented the SNI-induced decrease in intrinsic excitability (*Figure 5J*). Whereas PV neurons from control mice (*Pvalb*$^{Cre}$ mice injected with AAV9-CAG-DIO-eGFP-eIF4E-scrambled) exhibited reduced spiking activity and increased rheobase 4 weeks post-SNI compared to sham animals, PV neurons with reduced eIF4E showed no change in their excitability after nerve injury (*Figure 5J*: firing frequency; *Figure 5K*: rheobase). No change was found in membrane capacitance (*Figure 5L*), resting membrane potential (*Figure 5M*), and input resistance (*Figure 5N*).

We next investigated the effect of downregulating eIF4E-dependent translation in PV neurons on SNI-induced pain hypersensitivity using two complementary approaches. We first downregulated eIF4E by injecting AAV9-CAG-DIO-eGFP–eIF4E-shRNAmir into the lumbar dorsal horn parenchyma of *Pvalb*$^{Cre}$ mice 14 days prior to SNI. Unexpectedly, this manipulation did not attenuate mechanical hypersensitivity compared with mice injected with AAV9-CAG-DIO-eGFP–scrambled shRNA (*Figure 6A*). To confirm this finding, we used an alternative genetic approach. Specifically, we generated mice expressing a mutated, non-phosphorylatable form of 4E-BP1 (harboring threonine-to-alanine mutations at two key mTORC1 phosphorylation sites, amino acids 37 and 46; referred to as Tg-4EBP1mt) (*Tsai et al., 2015*) selectively in PV neurons (Tg-4EBP1mt; *Pvalb*$^{Cre}$). In these mice, the non-phosphorylatable 4E-BP1 binds to and inhibits eIF4E activity specifically in PV neurons. Consistent with the results of the AAV-eIF4E-shRNAmir-mediated knockdown experiment, we observed no alleviation of mechanical hypersensitivity in Tg-4EBP1mt; *Pvalb*$^{Cre}$ mice compared to control Tg-4EBP1mt

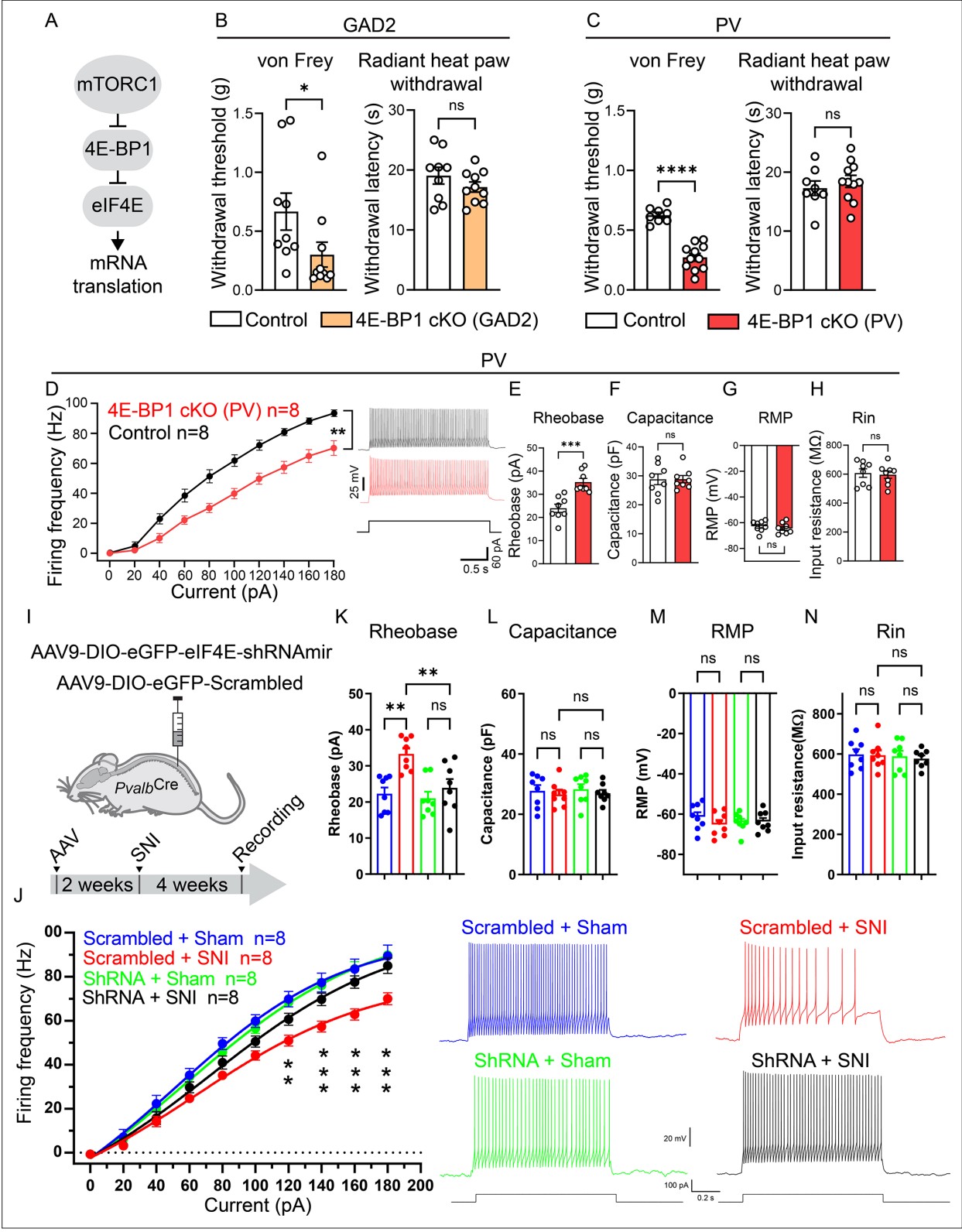

**Figure 5.** Activation of 4E-BP1-dependent translation in inhibitory neurons promotes mechanical hypersensitivity and contributes to reduced intrinsic excitability of PV neurons. (**A**) A schematic of mTORC1 pathway. Deletion of 4E-BP1 in GAD2 (**B**, 4E-BP1 cKO: *Eif4ebp1*fl/fl;*Gad2*Cre, Control: *Gad2*Cre, *n* = 9/10) and PV (**C**, 4E-BP1 cKO: *Eif4ebp1*fl/fl;*Pvalb*Cre, Control: *Pvalb*Cre, *n* = 8/11) neurons induces mechanical (50% withdrawal threshold) but not heat hypersensitivity. A comparable number of male and female mice was used in B and C. Recording from PV neurons in spinal cord slices (identified by the expression of L10a-eGFP) shows that the deletion of 4E-BP1 in PV neurons (4E-BP1 cKO: *Eif4ebp1*fl/fl: *L10a*-eGFP: *Pvalb*Cre, Control: *L10a*-eGFP:

*Figure 5 continued on next page*

*Figure 5 continued*

*Pvalb*[Cre], *n* = 8/8 female mice) induces a decrease in firing frequency (**D**) and an increase in rheobase (**E**). No change in membrane capacitance (**F**), resting membrane potential (RMP, **G**), and input resistance (Rin, **H**) was found. AAVs (AAV-CAG-DIO-eGFP-eIF4E-shRNAmir or AAV-CAG-DIO-EGFP-scrambled-shRNAmir) were injected into the parenchyma of the dorsal horn of *Pvalb*[Cre] female mice (illustration and time course are shown in **I**, *n* = 8/group), preventing the spared nerve injury (SNI)-induced decrease in PV neuron firing frequency (**J**) and elevation of rheobase (**K**). No changes were found in capacitance (**L**), RMP (**M**), and Rin (**N**). An unpaired two-tailed *t*-test was used in **B, C, E–H**. Two-way ANOVA followed by Tukey's post hoc comparison was used in J–N. Each data point represents an individual animal. Data are plotted as mean ± SEM. *p < 0.05, **p < 0.01, ***p < 0.001, ns – not significant.

The online version of this article includes the following figure supplement(s) for figure 5:

**Figure supplement 1.** Confirmation of 4E-BP1 and eIF4E downregulation.

animals (*Figure 6B*). Together, these results indicate that inhibiting eIF4E-dependent translation in PV neurons is sufficient to prevent the SNI-induced decrease in their intrinsic excitability, but is not sufficient to alleviate SNI-induced mechanical hypersensitivity.

Finally, we generated mice lacking 4E-BP1 in a broad population of excitatory Vglut2[+] neurons (*Eif4ebp1*[fl/fl]; *Slc17a6*[Cre]) (*Figure 6C*), as well as in a subpopulation of excitatory neurons defined by Tac1 (*Eif4ebp1*[fl/fl]; *Tac1*[Cre]) (*Figure 6D*). These two mouse lines exhibited no changes in mechanical or heat withdrawal thresholds, suggesting that increasing eIF4E-dependent translation in excitatory neurons does not confer pain hypersensitivity.

## Discussion

Previous studies in animal models of neuropathic pain have largely focused on changes in the transcriptome and were mostly limited to early time points after peripheral nerve injury. The growing realization of the important role of translational control in neuronal plasticity and pain sensitization, as well as the uncovering of distinct mechanisms underlying the development and maintenance phases of neuropathic pain (*Finnerup et al., 2021*; *Khoutorsky and Price, 2018*; *Gangadharan et al., 2022*; *Muralidharan et al., 2022*), prompted us to study gene expression at both transcriptional and translational levels during early and late time points after peripheral nerve injury. Unexpectedly, we discovered that gene expression during the maintenance phase in the spinal cord is substantially regulated at the level of mRNA translation. Furthermore, we demonstrated that downregulation of the key translation initiation factor eIF4E in the spinal cord, using cell type-non-specific ASO, leads to long-lasting alleviation of established pain hypersensitivity.

In the late phase of neuropathic pain, we found both transcriptional and translational changes in gene expression in DRG but greater translational changes in the spinal cord than transcriptional changes. Alterations in the DRG transcriptome are consistent with previous analyses in animal models (*Pokhilko et al., 2020*; *Parisien et al., 2019*; *Renthal et al., 2020*) and human DRG tissue from individuals with neuropathic pain, which revealed substantial transcriptional changes (*Ray et al., 2023*; *North et al., 2019*) accompanied by neuronal hyperexcitability (*North et al., 2019*). Gene expression datasets from human neuropathic spinal cord tissue are not yet available due to the paucity of spinal cord samples from individuals with neuropathic pain.

Altered translation at late time points after nerve injury might be linked to maladaptive spinal plasticity. eIF4E downregulation in the spinal cord alleviated pain hypersensitivity at the late, but not early, time point. Since translation is the predominant gene expression mechanism in the spinal cord at the late stage, it is conceivable that downregulation of eIF4E normalizes the translational landscape, thus correcting maladaptive plasticity underlying spinal hyperexcitability. In the early stage after nerve injury, modifications of existing proteins (e.g., via phosphorylation) and transcriptional changes play significant roles (*Finnerup et al., 2021*; *Colloca et al., 2017*), rendering suppression of translation less efficient.

In the maintenance phase of neuropathic pain, we observed greater translational changes, using FUNCAT and TRAP, in spinal inhibitory neurons compared to excitatory neurons. Moreover, enhancing 4E-BP1-dependent translation in GAD2[+] inhibitory and PV[+] neurons, but not excitatory neurons, induced mechanical hypersensitivity. Increasing translation was also sufficient to decrease the excitability of spinal PV interneurons, whereas suppressing translation in PV neurons prevented SNI-induced reduction in their excitability. Surprisingly, inhibiting eIF4E-dependent translation in PV neurons using

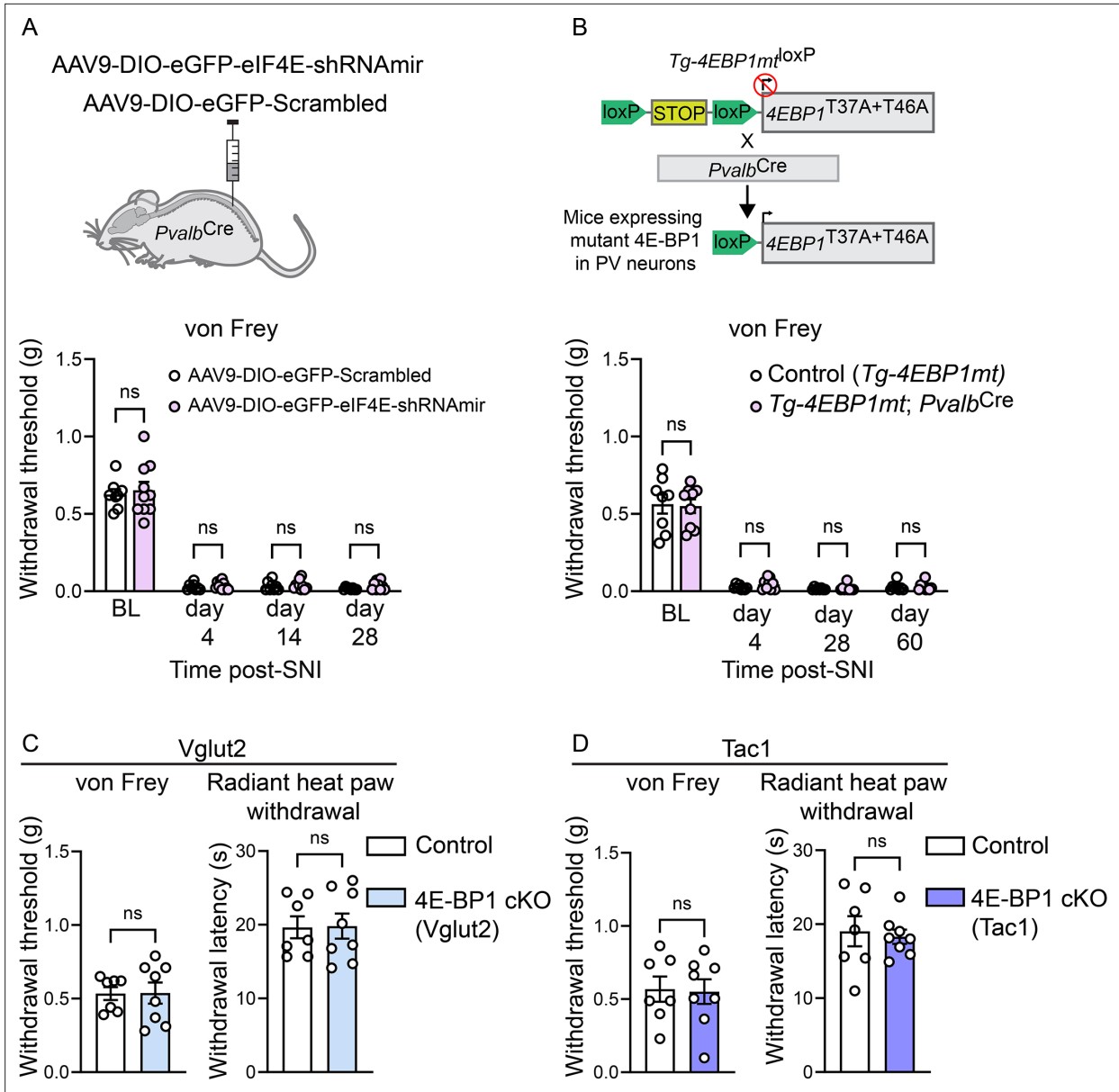

**Figure 6.** Effects of modulating eIF4E-dependent translation on pain hypersensitivity. (**A**) eIF4E was downregulated in PV neurons by intraspinal injection of AAV-CAG-DIO-eGFP-eIF4E-shRNAmir (and control AAV-CAG-DIO-EGFP-scrambled-shRNAmir) into the parenchyma of the dorsal horn of *Pvalb*Cre mice 2 weeks before spared nerve injury (SNI) (*n* = 10/8). (**B**) Mice expressing a mutated non-phosphorylatable 4E-BP1 in PV neurons (4EBP1mt;*Pvalb*Cre) and their controls (Tg-4EBP1mt) were subjected to SNI (*n* = 9/8). No reduction in SNI-induced mechanical hypersensitivity (von Frey, 50% withdrawal threshold) was observed in A or B. No changes were observed in mechanical (von Frey, 50% withdrawal threshold) and heat (radiant heat paw-withdrawal) thresholds in mice lacking 4E-BP1 in Vglut2 neurons (**C**, 4E-BP1 cKO: *Eif4ebp1*fl/fl; *Slc17a6*Cre, Control: *Slc17a6*Cre) or Tac1 neurons (**D**, 4E-BP1 cKO: *Eif4ebp1*fl/fl; *Tac1*Cre, Control: *Tac1*Cre, *n* = 7/8 mice). A comparable number of male and female mice was used in A–D. An unpaired two-tailed *t*-test. Data are plotted as mean ± SEM. ns – not significant.

two different approaches (eIF4E downregulation and expression of non-phosphorylatable 4E-BP1) did not alleviate SNI-induced mechanical hypersensitivity. There are several potential explanations for these results, including: (1) the presence of other mechanisms in PV neurons (e.g., a reduction in PV neuron synaptic output post-SNI mediated by retinoic acid receptor RARα; *Cao et al., 2022*) that are translation-independent; (2) the insufficiency of correcting reduced PV neuron excitability to alleviate hypersensitivity; and (3) an essential role for mRNA translation in other neuronal and/or non-neuronal cell types in neuropathic pain. Indeed, the alleviation of neuropathic pain hypersensitivity by eIF4E-ASO might be mediated by effects on multiple neuronal (e.g., excitatory and inhibitory

neurons) or non-neuronal (e.g., microglia, astrocytes, and immune cells) cell types, whereas reducing eIF4E-dependent translation solely in PV neurons is not sufficient to reverse hypersensitivity.

Spinal disinhibition, induced by peripheral nerve injury, plays a key role in central sensitization. Numerous neuronal and non-neuronal mechanisms contribute to this disinhibition, including: K$^+$–Cl$^-$ cotransporter (KCC2) downregulation causing elevation of intracellular chloride and the resulting weakening of inhibitory neurotransmission (*Coull et al., 2003*; *Coull et al., 2005*), preferential removal of inhibitory synapses by microglia (*Yousefpour et al., 2023*; *Kambrun et al., 2018*), and the modulation of the extracellular matrix (*Tansley et al., 2022*). In addition, peripheral nerve injury induces substantial plasticity in spinal PV neurons, leading to the reduction in their synaptic output as well as intrinsic excitability, thereby resulting in the engagement of myelinated low-threshold mechanoreceptive afferents in spinal nociceptive lamina I circuits (*Petitjean et al., 2015*; *Boyle et al., 2019*; *Cao et al., 2022*). Our data demonstrate that the reduction in PV neuron intrinsic excitability and spiking activity is mediated, at least partially, by translational activation. The exact molecular mechanisms underlying this form of plasticity, downstream of translation, remain unknown; however, the identification of translationally altered genes in TRAP analysis might facilitate their discovery.

Previous studies have shown that pharmacological targeting of mTORC1 can alleviate hypersensitivity in animal models of inflammation (*Khoutorsky and Price, 2018*; *Yousuf et al., 2021*; *Price et al., 2007*; *Xu et al., 2011*; *Asante et al., 2009*) and inhibition of mTORC1 shortly before or after nerve injury transiently alleviates pain hypersensitivity (*Géranton et al., 2009*; *Khoutorsky and Price, 2018*). These effects could be mediated through the downregulation of translation in DRG neurons, or alternatively, via inhibiting translation-independent functions of mTORC1 such as lipid biogenesis, regulation of mitochondrial functions, and autophagy, which are all implicated in neuropathic pain (*Liao et al., 2022*; *Miller et al., 2020*; *Roh et al., 2020*; *Silva Santos Ribeiro et al., 2022*). Targeting eIF4E in the spinal cord via eIF4E-ASO does not affect DRG neurons and specifically inhibits the translational control mechanism, without affecting other functions of mTORC1.

Our study has several limitations. eIF4E-ASO was administered via the i.c.v. route to specifically target the CNS without affecting DRGs. Although we showed unaltered locomotor function using rotarod and open field tests and measured von Frey reflexive responses, we cannot completely rule out supraspinal effects of eIF4E-ASO. Notably, eIF4E-ASO is not cell type-specific and therefore does not allow conclusions about which cell type(s) mediate the pain-alleviating effects.

We used both sexes in behavioral experiments but only female mice in all other experiments. Therefore, future translational profiling studies in neuropathic pain should be extended to males. We used slightly different time points for Ribo-seq (day 63 for the late phase) and TRAP (day 60 for the late phase) experiments. Since we did not make a direct comparison between the two approaches, we believe that these differences in time points do not affect the conclusions of the study. Finally, we used cKO mice in which Cre expression begins either during embryonic or early postnatal stages (*Tac1*$^{Cre}$, *Gad2*$^{Cre}$, and *Vglut2*$^{Cre}$). This may potentially lead to developmental alterations that could confound the conclusions derived from these experiments.

In summary, our study provides a characterization of the translational landscape at early and late time points of neuropathic pain and in a subset of excitatory and inhibitory neurons. We identified the substantial role of spinal translation during the late stage of neuropathic pain and revealed that ASO-mediated downregulation of spinal translation provided a long-lasting alleviation of established pain hypersensitivity. These findings enhance our understanding of the cell type- and phase-specific mechanisms underlying neuropathic pain, raising the possibility for the development of targeted therapeutics.

## Materials and methods
### Animals and housing conditions
C57BL/6 mice were purchased from Charles River Laboratories, Inc (St. Constant, Quebec, Canada) at 6–7 weeks of age. On arrival at the in-house animal facility, the mice were placed in groups of 5 animals per cage with food and water ad libitum under a 12:12-hr light/dark cycle (light period from 07:00 to 19:00 hr) with ambient temperature (22°C) and humidity maintained at 40%. *Eif4ebp1*$^{fl/fl}$ mice (*Aguilar-Valles et al., 2021*) were crossed with *Gad2*$^{Cre}$ (The Jackson Laboratory, stock #010802) to generate *Eif4ebp1*$^{fl/fl}$;*Gad2*$^{Cre}$ animals, *Pvalb*$^{Cre}$ (Jackson Laboratory, stock #008069) to generate

*Eif4ebp1*<sup>fl/fl</sup>;*Pvalb*<sup>Cre</sup> animals, *Tac1*<sup>Cre</sup> (The Jackson Laboratory, stock #021877) to generate *Eif4ebp1*<sup>fl/fl</sup>;*Tac1*<sup>Cre</sup> animals, and *Slc17a6*<sup>Cre</sup> (The Jackson Laboratory, stock #028863) to generate *Eif4ebp1*<sup>fl/fl</sup>;*Slc17a6*<sup>Cre</sup> animals (corresponding Cre lines were used as controls). *Tg-4EBP1mt* mice (Jackson Laboratory, stock #029735) *Tsai et al., 2015* were crossed with *Pvalb*<sup>Cre</sup> (Jackson Laboratory, stock #008069) to generate *Tg-4EBP1mt;Pvalb*<sup>Cre</sup> animals. *L10a*-eGFP; *Tac1*<sup>Cre</sup> and *L10a*-eGFP; *Gad2*<sup>Cre</sup> mice were generated by crossing *L10a*-eGFP mice (*Sanz et al., 2009*) with the corresponding Cre mouse lines. We also generated *Eif4ebp1*<sup>fl/fl</sup>: *L10a*-eGFP: *Pvalb*<sup>Cre</sup>, *Eif4ebp1*<sup>fl/fl</sup>: *L10a*-eGFP: *Gad2*<sup>Cre</sup>, and *Eif4ebp1*<sup>fl/fl</sup>: *L10a*-eGFP: *Tac1*<sup>Cre</sup> mice and their controls (no *Eif4ebp1*<sup>fl/fl</sup>) for confirmation of 4E-BP1 deletion experiments and recording from spinal PV neurons (*Figure 5D–H*). Sample sizes were determined based on similar previous studies in the field. Female mice were used in Ribo-seq (*Figure 1*), TRAP (*Figure 4*), FUNCAT (*Figure 3*), and electrophysiology experiments. All experiments were performed and analyzed by an experimenter blind to genotypes and treatments. Both sexes (with comparable numbers of males and females) were used in all behavioral experiments. No main effects of sex were observed; therefore, data were pooled for all reported analyses. All procedures complied with the Canadian Council on Animal Care guidelines and were approved by the McGill University's Downtown Animal Care Committee (protocol #7869).

## Spared nerve injury

Mice were anesthetized under 4% isoflurane for induction and 2% for maintenance. The mice were placed on a heated (36–37°C) surgical bed during the surgical procedure. The sciatic nerve was exposed by making an incision in the upper thigh and cutting through the femoris muscle. The tibial and common peroneal branches were ligated with 7.0 silk (Covidien, S-1768K), and a 2- to 4-mm section of the nerve below the ligation was removed using micro self-opening scissors and forceps, leaving the sural nerve fully intact. The muscle and skin were sutured using 6.0 Vicryl (Ethicon, J489G). All sham surgeries featured incisions to the thigh and cutting of the femoris muscle; however, the sciatic nerve was left untouched and intact. Mice returned to their cage placed on a heated (36–37°C) surface for recovery.

## Western blotting

Mice were decapitated 2 weeks post-ASO injection. The lumbar section of the spinal cord and the DRGs were extracted and immediately homogenized in a homogenization buffer 200 mM HEPES, 50 mM NaCl, 10% glycerol, 1% Triton X-100, 1 mM EDTA, 50 mM NaF, 2 mM Na3VO4, 25 mM β-glycerophosphate, and EDTA-free complete ULTRA tablets (Roche, Indianapolis, IN) before being centrifuged at 14,000 rpm for 15 min at 4°C to obtain a supernatant. Bradford protein assay was used to measure the protein concentration of the lysates, followed by loading 30 μg of the lysates on a 12% SDS–PAGE gel, and ran at a constant current (0.03 A/gel). The gel was transferred to a nitrocellulose membrane overnight on ice at 20 mV. The membrane was blocked (5% milk or BSA in TBS-T) for 1 hr and then incubated in primary antibodies overnight at 4°C. Afterwards, the membrane was washed three times and incubated in HRP-conjugated secondary antibody at room temperature. The membrane was then washed three times, and an Enhanced Chemiluminescent (ECL) reagent was used to enhance the signal before visualizing it using a ChemiDoc Imaging System (Bio-Rad). Primary antibodies were eIF4E BD (Biosciences, Cat# 610270, 1:1000), and beta-tubulin (Cell Signaling, Cat# 2146S, 1:1000). Secondary antibodies were anti-Rabbit IgG – Horseradish Peroxidase antibody (GE Healthcare, Cat# NA9340 RRID:AB_772191), and sheep anti-Mouse IgG – Horseradish Peroxidase antibody (GE Healthcare, Cat # NA931; RRID:AB_772210).

## Fluorescent noncanonical amino acid tagging

Female mice, aged 8–10 weeks, were fed a methionine-free diet (Envigo RMS Inc, TD.110208) for 1 week, followed by an intraperitoneal injection of AHA (100 μg/g body weight, i.p., Click-IT AHA [L-azidohomoalanine], Cat No. C10102, Thermo Fisher Scientific). After 3 hr, mice were anesthetized and perfused transcardially with 4% paraformaldehyde (PFA) in phosphate-buffered saline (PBS), pH 7.4. The L4 and L5 lumbar sections of the spinal cord were extracted and kept at 4°C in PFA overnight. Tissue was then cut into 30 μm sections and washed three times (10-min increments) in PBST (0.2% Triton X-100 in PBS) before being blocked overnight in a solution composed of 10% normal goat serum, 0.5% Triton-X100, and 5% sucrose in PBS. Afterward, click chemistry was performed on

the sections overnight in a click buffer, consisting of 200 μM triazole ligand, 400 μM TCEP, 2 μM fluo-rescent Alexa Fluor 555 alkyne (Alexa Fluor 555 Alkyne, Cat No. A20013, Thermo Fisher Scientific), and 200 μM $CuSO_4$ in PBS. The sections were washed followed by immunohistochemistry (described below). Quantification (25 neurons were quantified per mouse) was performed as described below for immunohistochemistry. Anisomycin (100 mg/kg, A9789, Sigma-Aldrich) was injected intraperitoneally 1 hr before AHA administration.

## Immunohistochemistry

Mice were anesthetized and perfused with 4% PFA in PBS, pH 7.4, and spinal cords were extracted and left overnight in PFA at 4°C. Spinal cords were then transversely cut into 30 μm sections followed by three washes (10 min each) using 0.2% Triton X-100 in PBS. After washing, sections were blocked using a solution consisting of 5% normal donkey and 5% normal goat serum and 0.2% Triton X-100 in PBS for 1 hr. After blocking, tissue was incubated overnight in primary antibodies diluted in PBS. Sections were washed three times in PBS and incubated in the corresponding secondary antibody diluted in PBS for 2 hr.

Primary antibodies for immunohistochemistry were: NeuN (1:1000, mouse, Millipore, MAB377), 4E-BP1 (1:200, rabbit, #2855S, Cell Signaling and Technology Laboratories), Pax2 (1:500, goat, Novusbio, AF3364), and GFP (1:1000, chicken, Abcam, ab13970). Secondary antibodies were Goat anti-rabbit Alexa Fluor 568 (1:500, Thermo Fisher Scientific, A-11011), Donkey anti-goat Alexa Fluor 647 (Thermo Fisher Scientific, A-21447), Goat anti-mouse Alexa Fluor 488 (Thermo Fisher Scientific, A-21042), and Alexa Fluor 555 alkyne (Alexa Fluor 555 Alkyne, Cat No. A20013, Thermo Fisher Scientific). Antibodies against eIF4E (1:500, mouse, Santa Cruz, SC-271480) and GFP (1:1000, chicken, Abcam, ab13970) were used to confirm knockdown of eIF4E in $PV^+$ cells. Secondary antibodies were Goat anti-mouse Alexa Fluor 488 (Thermo Fisher Scientific, A-21042) and Goat anti-chicken Alexa Fluor 647 (Thermo Fisher Scientific, A-21449).

After three washes in PBS, the sections were mounted and imaged using a Zeiss confocal microscope (LSM 880, 63X/1.40 Oil DIC f/ELYRA objective) equipped with Argon, DPSS, and HeNe lasers. Alexa Fluor 488 was excited using the Argon laser and detected with an emission window of 507–551 nm. Alexa Fluor 555/568 was excited using the DPSS 561 nm laser and detected with an emission window of 578–631 nm. Alexa Fluor 647 was excited using the HeNe 633 nm laser and detected with an emission window of 654–690 nm. Images were acquired using identical settings across experimental groups. Integrated density of AHA signal was measured within $Pax2^+$ cells for inhibitory neurons, or $Pax2^-/NeuN^+$ cells for excitatory neurons in the superficial dorsal horn (laminae I–III, defined using a lamina overlay based on NeuN staining) and quantified using ImageJ on maximum intensity projection images. To confirm the knockout of 4E-BP1 signal, integrated density of 4E-BP1 signal was measured in eGFP⁺ neurons using *Eif4ebp1*ᶠˡ/ᶠˡ: *L10a*-eGFP: *Pv*ᶜʳᵉ, *Eif4ebp1*ᶠˡ/ᶠˡ: *L10a*-eGFP: *Gad2*ᶜʳᵉ, and *Eif4ebp1*ᶠˡ/ᶠˡ: *L10a*-eGFP; *Tac1*ᶜʳᵉ transgenic mice. Corresponding mouse lines without *Eif4ebp1*ᶠˡ/ᶠˡ were used as controls. Background noise was subtracted from the final calculations in all experiments. For all image quantification experiments, three sections were analyzed per mouse, with three images acquired from each section (nine images per mouse). The integrated density of AHA, eIF4E, or 4E-BP1 signals in the somata of 25 neurons per mouse was quantified using ImageJ on maximum-intensity projection images (generated using the same parameters and number of optical sections), and the values were averaged per mouse and presented.

## Translating ribosomal affinity purification

SNI and Sham surgeries were performed on 8- to 10-week-old female (*L10a*-eGFP; *Tac1*ᶜʳᵉ and *L10a*-eGFP; *Gad2*ᶜʳᵉ) mice 4 or 60 days prior to tissue extraction. TRAP-sequencing was performed as previously described (*Megat et al., 2019*; *Sanz et al., 2009*). Mice were decapitated and the lumbar section of the spinal cord was extracted under cold and RNase-free conditions and transferred to ice-cold dissection buffer (1X HBSS, 2.5 mM HEPES-NaOH [pH 7.4], 35 mM Glucose, 5 mM $MgCl_2$; 100 μg/ml cycloheximide and 0.2 mg/ml emetine were added just before use). Next, the extracted spinal cord tissues were homogenized in lysis buffer (20 mM HEPES-NaOH [pH 7.4], 12 mM $MgCl_2$ and 150 mM KCl in RNAse free water; 0.5 mM DTT, 1 μl/ml Protease Inhibitors Cocktail [Roche], 100 μg/ml cycloheximide, 20 μg/ml Emetine, 40 U/ml RNasin [Promega] and 2 U/ml TURBO DNase [Invitrogen] were added immediately before use) using Minilys Personal High Power Tissue Homogenizer (Bertin

Technologies) on medium speed for 10 s for a total of eight times with 10-s intervals (incubation on ice) in a cold room at 4°C. A post-nuclear fraction was generated from spinal cord homogenates by centrifuging at 2000 × $g$ for 5 min at 4°C. The supernatant was collected and NP-40 (AG Scientific) and DHPC (Avanti Polar lipids) were added at a final concentration of 1% and incubated on ice for 5 min. Afterward, post-mitochondrial fractions were generated by centrifuging samples at 15,000 × $g$ for 10 min. A 200-µl aliquot was taken from the post-mitochondrial fraction and used as the input, and the remaining fraction was incubated overnight with protein washed G-coated Dynabeads (Invitrogen) bound to 50 µg of anti-GFP antibodies (HtzGFP-19F7 and HtzGFP-19C8 antibodies were acquired from Sloan Memorial Kettering Centre) on an end-over-end mixer. On the following day, the beads were washed four times with a high salt buffer (20 mM HEPES-NaOH [pH 7.4], 12 mM $MgCl_2$ and 0.35 M KCl, 1% NP-40 [AG Scientific] in RNAse free water; 100 µg/ml cycloheximide and 0.5 mM DTT were added just before use) to collect the IP fraction. After the removal of the final wash, RNA was extracted by incubating 300 µl of Trizol to the IP fraction and 600 µl to the input fraction for 10 min at room temperature. Equal amounts of ethanol were added to each of the samples before eluting the RNA using a Direct-zol RNA kit (Zymo Research) using manufacturer's protocol. RNA yields were quantified using a NanoDrop Spectrophotometer ND-1000 (NanoDrop Technologies, Inc) and RNA quality was determined by a 2100 Bioanalyzer (Agilent Technologies).

## Library generation and sequencing

Sequencing was performed on IP and corresponding input fractions. Three lumbar spinal cords were pooled per replicate, with a total of three replicates per group. Groups were labeled as follows: day 4 SNI Tac1 or GAD2, day 4 Sham Tac1 or GAD2, day 60 SNI Tac1 or GAD2, and day 60 Sham Tac1 or GAD2. Total RNA quality assessment, library generation, library quality check, and sequencing were carried out at the Génome Québec (Montreal).

Total RNA was quantified, and its integrity was assessed on a LabChip GXII (PerkinElmer) instrument. rRNA was depleted from 70 ng of total RNA using QIAseq FastSelect (Human/Mouse/Rat 96rxns). cDNA synthesis was achieved with the NEBNext RNA First Strand Synthesis and NEBNext Ultra Directional RNA Second Strand Synthesis Modules (New England BioLabs). The remaining steps of library preparation were performed using the NEBNext Ultra II DNA Library Prep Kit for Illumina (New England BioLabs). Adapters and PCR primers were purchased from New England BioLabs. Libraries were quantified using the KAPA Library Quantification Kits – Complete kit (Universal) (Kapa Biosystems). The average size fragment was determined using a LabChip GXII (PerkinElmer) instrument.

The libraries were normalized and pooled and then denatured in 0.05 N NaOH and neutralized using HT1 buffer. The pool was loaded at 175 pM on an Illumina NovaSeq S4 lane using Xp protocol as per the manufacturer's recommendations. The run was performed for 2 × 100 cycles (paired-end mode). A phiX library was used as a control and mixed with libraries at 1% level. Base calling was performed with RTA v3.4.4. Program bcl2fastq2 v2.20 was then used to demultiplex samples and generate fastq reads. mRNA library preparation and sequencing were done at Genome Quebec.

## TRAP bioinformatics and statistical analysis

### Mapping and TPM quantification

Quality of FASTQ files was checked using FastQC (Babraham Bioinformatics, https://www.bioinformatics.babraham.ac.uk/projects/fastqc/). Phred scores, per-base sequence, and duplication levels were analyzed. Reads were soft-clipped (12 bases per read) to ignore adapters and low-quality bases during alignment using STAR v2.7.6 with the GRCm39 mouse reference genome (Gencode release M31, primary assembly) (*Dobin et al., 2013*). Reads were also sorted with STAR and then deduplicated with sambamba v0.8.2 (*Tarasov et al., 2015*). StringTie v2.2.1 was used to obtain Transcript Per Million (TPM) values for each gene of all samples (*Pertea et al., 2015*). Non-coding and mitochondrial genes were removed and TPM values for coding genes were re-normalized to sum to 1 million before performing downstream analysis.

### Order statistics and re-normalization of expression data

Downstream analysis of TRAP datasets was done as previously described (*Sanz et al., 2009*; *Wong et al., 2023*). Each transcriptome sample (INPUT) had consistently expressed genes identified by calculating percentile ranks for each coding gene. We identified between 14,763 and 14,812 genes,

dependent on condition, that were above the 30th percentile in each INPUT sample. Quantile normalization was performed based on the set of all coding genes. IP (translatome) analysis was performed on the consistently transcriptome-expressed genes. To identify consistently expressed genes in the translatome samples, the percentile ranks of TPM were calculated for each of the consistently transcriptome-expressed coding genes of each sample. We identified between 13,140–13,191 out of 14,763–14,812 genes, dependent on condition, that are consistently detected in the translatome based on whether their expression was on or above the 10th percentile in each IP sample.

### Differential expression analysis

Differential expression (DE) analysis was done as previously described by *Tavares-Ferreira et al., 2022*. $Log_2$-fold change was calculated based upon median TPM values for each transcriptome-expressed and translatome-expressed coding gene in the INPUT and IP samples, respectively. Strictly standardized mean difference (SSMD) was used to reveal genes with systematically altered expression percentile ranks between SHAM and SNI mice of the same cell type and time point. SSMD is the difference of means controlled by the variance of the sample measurements. SSMD measured the effect size to control for within-group variability. Differentially expressed genes were determined between Sham and SNI of the same cell type and time point by calculating the Bhattacharyya distance (*Zhang et al., 2009*). This measure is used to calculate the amount of overlap in the area under the curve of the two sample distributions (corresponding to each group). BD compares the distribution of gene relative to abundance (in TPMs). The Bhattacharyya coefficient BC(Q)i ranges between 0 (for totally non-overlapping distributions) and 1 (for completely identical distributions) and is derived from the Bhattacharyya distance. In our analysis, we used a modified form of the Bhattacharyya coefficient that ranges between 0 (for completely identical distributions) and +1 or –1 (for totally non-overlapping distributions, sign defined by the log-fold change value). DE genes were identified if the absolute value of SSMD was higher or equal to 0.97, the absolute value of BC was higher or equal to 0.5, and fold change higher or equal to 1.33. Coding for bioinformatics analysis and data visualization was done in Python (version 3.7 with Anaconda distribution).

## Behavioral pain studies

### von Frey

Mice were habituated for 1 hr in individual transparent Plexiglas cubicles (5 cm × 8.5 cm × 6 cm) placed on a perforated steel floor. Mice were then tested by applying calibrated nylon monofilaments perpendicular to the surface of the hind paw for three seconds. Withdrawal of the mouse's foot before the monofilament buckled was considered a positive response. The up-down method of Dixon was used to estimate the 50% withdrawal threshold (average of two measurements separated by at least 30 min) (*Chaplan et al., 1994*).

### Radiant heat paw-withdrawal (Hargreaves') assay

Mice were habituated for 1 hr in individual transparent Plexiglas cubicles (5 cm × 8.5 cm × 6 cm) placed on a transparent glass floor. During testing, a high-intensity light source was applied to the surface of the hind paw. Intensity was set at 20% (of the maximum) using the IITC model 390. Latency of the hind paw withdrawal was measured. The cut-off for a response was 40 s. Hind paws were measured twice, separated by at least 30 min.

### Mouse Grimace Scale

The MGS was adopted from a previous publication (*Langford et al., 2010*). Mice were habituated in custom-made Plexiglas cubicles (5.3 cm × 8.5 cm × 3.6 cm) for 1 hr. After habituation, mice were recorded (Sony Digital Camcorder HDR-PJ430V) for 1 hr. One photo was chosen from every 3 min period for a total of 20 pictures. These pictures were then randomized, and the coder was blinded to the groups before they were analyzed to give each mouse their mean score. Scoring was based off five facial features (action units) including orbital tightening, nose bulge, cheek bulge, ear position, and whisker change. Scores ranged from 0 to 2, with 0 being no evidence of the action unit present, 1 moderate evidence of the action unit, and 2 obvious evidence of the action unit. The final MGS score was given to each mouse based upon averaging the intensity ratings for all five action units.

### Rotarod test

Motor coordination was evaluated using the IITC Life Science Rotarod apparatus. Mice were first habituated to the task by balancing on a rotating rod set at a constant speed of 5 rpm for 1 min. This habituation session was conducted three times, with 5-min intervals between trials. During the test session, the rod initially rotated at 5 rpm and gradually accelerated at a rate of 0.2 rpm/s over a 5-min period. The latency to fall was automatically recorded when the animal dropped onto the sensor located beneath the rod. Each mouse underwent three test trials, separated by 10-min intervals. Both the time to fall (s) and the rotational speed at the moment of falling (rpm) were measured.

### Open field test

General locomotor activity was assessed by allowing mice to freely explore an open field arena (44 cm × 44 cm) for 5 min. Movement was recorded and analyzed using an automated video-tracking system (Ethovision 5.0, Noldus) to determine total distance traveled and average velocity.

### I.c.v. stereotaxic injection

Mice were deeply anesthetized (induced with 5% isoflurane and maintained on 2% isoflurane) and their head was secured using ear bars in a stereotaxic frame (Kopf). The head was shaved, and an incision was made to expose the skull. Aiming occurred via coordinates relative to bregma (anterior/posterior (AP): –0.5 mm, medial/lateral (ML): 1 mm, and dorsal/ventral (DV): –2.2 mm), and the skull was drilled for injection of the ASO into the lateral ventricles (i.c.v.). Five µl of eIF4E-ASO or scrambled ASO was injected using a 10-µl Hamilton microsyringe mounted on a perfusion pump. The perfusion rate was set to 0.5 µl/min, and the needle stayed in place for an additional 5 min to prevent leakage. The timeline of ASO administration, SNI, and behavioral testing is shown in *Figure 2*.

### eIF4E antisense oligonucleotide

eIF4E-ASO targeting mouse *Eif4e* mRNA and Control scrambled ASO were developed and synthesized by Ionis Pharmaceuticals, see *Supplementary file 2* for sequences. Each ASO consists of 5 nucleotides on the 5' and 3' ends of the ASO with a 2'-O-methoxyethyl modification, and a central 10-base DNA 'gap'. ASO was diluted in PBS to a concentration of 100 mg/ml and was delivered i.c.v. at a single dose of 100 mg/kg.

### Intraspinal AAV injections

Eight- to ten-week-old *Pv*Cre mice were injected with AAV-CAG-DIO-eGFP-eIF4E-shRNAmir or AAV-CAG-DIO-EGFP-scrambled-shRNAmir using a minimally invasive (non-laminectomy) technique. Mice were deeply anesthetized (induced with 5% isoflurane and maintained on 2% isoflurane) and steel clamps were attached to the vertebral column. An incision was made in the skin and muscle at T12–L3, removing the muscle from the space between the T13 and L1 vertebrae. A glass electrode was inserted 250 µm into the spinal dorsal horn. A total of 500 nl of AAV-shRNA was injected over a 10-min period (David Kopf instruments, 99236B) followed by suturing of the skin using 6-0 Vicryl silk (Ethicon, J489G).

### AAV9-shRNAmir cloning and preparation

The microRNA-adapted short hairpin RNAs (shRNAmir) packaged in adeno-associated virus (AAV9-CAG-DIO-eGFP-eIF4E-shRNAmir and AAV9-CAG-DIO-eGFP-scrambled-shRNAmir) were prepared by Vector Biolabs. The validated sequence targeting mouse eIF4E was TCCAGTTGTCTTAATTTAAG TCAGTCAGTGGCCAAAACTTAAATTACTAGACAACTGGACAG, and the scrambled sequence used as a control was GAAATGTACTGCGCGTGGAGACGTTTTGGCCACTGACTGACGTCTCCACGCA GTACATTTCAG.

### Electrophysiology

#### Tissue preparation

Six- to eight-week-old mice were anesthetized with 200 mg/kg tribromoethanol (Avertin, i.p.) and cardiac perfused with 4°C NMDG-ACSF solution containing (in mM): 92 NMDG, 2.5 KCl, 1.2 $NaH_2PO_4$, 30 $NaHCO_3$, 20 HEPES, 25 glucose, 5 sodium ascorbate, 2 thiourea, 3 sodium pyruvate, 10 $MgSO_4$,

0.5 CaCl$_2$ (pH = 7.3–7.4, 300–310 mOsm). Following the cardiac perfusion, the vertebral column was rapidly removed and placed in the same oxygenated NMDG-ACSF solution described above. The vertebrae were removed with microforceps and microscissors, under a Zeiss Stemi 305 stereo microscope, and the dorsal/ventral roots were clipped close to the DRG. The spinal cord of the lumbar region was carefully peeled from the dura mater and superfluous roots and glued to a 2% agar block with the dorsal side up, and then embedded in 3% low melting point agarose. Transverse parasagittal slices (250 µm thick) were made using a Vibratome (Leica VT1200). Slices were incubated at room temperature for 30–45 min in oxygenated recovery solution containing (in mM): 92 NaCl, 2.5 KCl, 1.2 NaH$_2$PO$_4$, 30 NaHCO$_3$, 20 HEPES, 25 glucose, 5 sodium Aacorbate, 2 thiourea, 3 sodium pyruvate, 2 MgSO$_4$, 2 CaCl$_2$ (pH = 7.3–7.4, 300–310 mOsm).

## Whole-cell recordings

During recording, the spinal cord preparation was kept submerged and perfused with ACSF containing (mM): 125 NaCl, 25 NaHCO$_3$, 1.25 KCl, 1.25 KH$_2$PO$_4$, 1.5 MgCl$_2$, 1.5 CaCl$_2$, and 16 glucose and saturated with 95% O$_2$–5% CO$_2$, the temperature was kept constant (within ±0.5°C), at 30°C.

Fluorescent-labeled PV neurons at the laminae II and III in the dorsal region of the spinal cord slices were visually identified under the fluorescent microscope (Zeiss axiocam 506). To measure the intrinsic membrane properties and neuronal excitability, whole-cell recordings were performed in current-clamp mode with an increment step of 20 pA (0–180 pA), 1- to 2-s current injections. The patch pipettes were filled with intracellular solution containing (in mM): 140 K-gluconate, 2.5 MgCl$_2$, 10 HEPES, 2 Na$_2$-ATP, 0.5 Na$_2$-GTP, and 0.5 EGTA (pH 7.3, 295–305 mOsm), and recordings were excluded if the RMP was more positive than −50 mV or series resistance was >25 MΩ.

The data were acquired with pCLAMP 11.0 (Molecular Devices) at a sampling rate of 10–20 kHz and were measured and plotted with pCLAMP 11.0 and GraphPad Prism 9.0. Results are reported as mean ± SEM. Statistical analysis of the data was performed using unpaired $t$-test or one-way ANOVA, followed by Tukey's multiple comparison test. Statistical significance was set at p < 0.05.

## Ribo-seq

### Tissue extraction

On day 4 and 63 post-SNI or sham surgery, mice were euthanized by isoflurane anesthesia followed by decapitation. Immediately after, the mouse was held vertically, with the rostral end facing down, allowing blood to drain from the trunk for about 15–20 s. Then the mouse body was promptly placed and secured on a bed of dry ice using duct tape. The spinal column was carefully cut open to expose the spinal cord and the DRGs, which were doused with RNA*later* (Invitrogen, AM7020). The lumbar DRGs were extracted from the L3, L4, and L5 vertebrae levels and the corresponding section of the spinal cord was extracted from the T12, T13, and L1 vertebrae levels. The extracted tissues were quickly placed in pre-chilled, RNase-free microcentrifuge tubes (Ambion) and snap frozen by submerging the tubes in liquid nitrogen for a few seconds. The tubes were then stored in a –80°C freezer until tissue from all animals was collected. Tissue from 15 animals was pooled for each ribosome footprinting (Ribo-seq) replicate.

### Ribosome footprinting and RNA-seq

All consumables and solutions used were certified RNase free by the manufacturer. Bench-top, pipettes, centrifuge rotors, gel tanks, and glass Dounce homogenizers were cleaned with RNaseZap (Invitrogen, AM9780) as per the manufacturer's instructions.

### Tissue homogenization

Tissue homogenization, RNA and footprint extraction, and library preparation were carried out according to the protocol described by *Ingolia et al., 2012* with minor modifications as outlined in *Uttam et al., 2018*.

Briefly, frozen tissue (DRGs or spinal cord, pooled from 15 animals per replicate) was homogenized in 800 µl lysis buffer using pre-chilled 2 ml glass Dounce Homogenizers, performing 30 strokes each with pestle A followed by pestle B. The tissue lysate was collected in a microcentrifuge tube and centrifuged at 16,000 × $g$ for 15 min at 4°C. The total crude RNA content in the supernatant was

determined using a NanoDrop 1000. A fraction of tissue lysate containing 100 µg total crude RNA was reserved for mRNA-seq and the remaining was used for ribosome footprinting.

## Nuclease footprinting

It was ensured that one complete set of replicates had the same amount of crude total RNA to start with. The RNase I treatment was carried out as described by *Ingolia et al., 2012* for 45 min at 4°C using 5 µl of RNase I (Ambion, AM2295) per 250 µg of crude total RNA and quenched by adding 20 µl of SUPERase-In (Ambion, AM2694) per 5 µl of RNase I used.

## Recovery of ribosome-associated footprints

Ribosome-associated footprints were pelleted by ultra-centrifuging the RNAse I digestion mix (final volume 540 µl) layered on top of a 660-µl sucrose cushion at 71,000 rpm at 4°C in a Beckman Coulter TLA-120 rotor. The ribosomal pellet was resuspended in 600 µl of 10 mM Tris pH 7 (prepared from 1 M Tris pH 7, Ambion) and stored at –80°C until all samples for each tissue type were processed until this step.

RNA was purified from the resuspended ribosomal pellet using the Hot-Phenol RNA Extraction method and precipitated with isopropanol. For the Hot-Phenol RNA extraction, the resuspended ribosomal pellet was brought to a final volume of 700 µl with 10 mM Tris pH 7 and supplemented with 40 µl of 20% SDS followed by heated incubation at 65°C for 1 min at 1400 rpm in a thermomixer. The heated sample-SDS mix was equally split into two microcentrifuge tubes, containing hot acidic phenol (heated to 65°C) and incubated at 65°C, 1400 rpm for 5 min in the thermomixer. Subsequently, the tubes were chilled on ice for 5 min and then centrifuged at $2000 \times g$ for 3 min at 4°C to obtain a top aqueous phase, which was promptly collected in a fresh tube. Seven hundred µl of acidic phenol was added to the aqueous phase and incubated at 25°C, 14,000 rpm in a thermomixer followed by centrifugation to again obtain the top aqueous phase. The top aqueous phase was promptly collected in a fresh tube, to which 600 µl of chloroform was added and mixed by vortexing for 1 min at room temperature. The tubes were then centrifuged to recover the top aqueous phase, which was transferred to a fresh tube. The volume of the aqueous phase was estimated using a pipette and 1/9th volume of 3 M sodium acetate pH 5.5 (to a final concentration of at least 0.3 M sodium acetate) and 2 µl of 15 mg/ml Glycoblue were added. The tube was vortexed briefly, and 1 volume of pre-chilled isopropanol was added, followed by overnight storage at –80°C overnight to aid precipitation. The next morning, the tubes were centrifuged at $20,000 \times g$ at 4°C for 30 min to pellet the precipitated RNA. The supernatant was discarded, and the pellet was washed with 750 µl of ice-cold 80% ethanol, followed by centrifugation at $20,000 \times g$ at 4°C for 5 min. The supernatant was discarded, and the pellet was air-dried for up to 5 min before resuspending in 21 µl of 10 mM Tris, pH 7. One µl of the purified RNA was used to assess the RNA concentration on NanoDrop 1000.

## Purification of footprint fragments and dephosphorylation

Purification of footprint fragments from the ribosome-associated footprint complex and subsequent dephosphorylation was carried out as described in steps 18–29 of *Ingolia et al., 2012*. In step 25, we selected the Rapid Gel Extraction method to extract RNA from polyacrylamide gels. The resultant RNA pellet (dephosphorylated footprints) was resuspended in 9 µl of 10 mM Tris pH 7. One µl of the RNA was used to assess the size and concentration of the footprints obtained on the Agilent Bioanalyzer using a Small RNA Bioanalyzer Kit and the manufacturer's protocol.

## rRNA depletion

Half of the obtained dephosphorylated footprint RNA was processed further for rRNA depletion using Ribo-Zero Gold rRNA Removal Kit (Human/Mouse/Rat) (Illumina, MRZG12324). For each reaction, 90 µl of magnetic beads were transferred to a RNase-free tube and washed two times (1 min each) with an equal volume of RNase-Free water (Ambion, 4387936) using a magnetic stand and resuspended in 35 µl of resuspension solution. To 4 µl of dephosphorylated RNA samples, 12 µl of RNase-free water, 2 µl of Ribo-Zero Reaction Buffer, and 2 µl of Ribo-Zero rRNA Removal Solution was added, mixed by pipetting, and incubated at 68°C for 10 min, followed by a 5-min incubation at room temperature. The treated RNA samples were then transferred to a 1.5-ml microcentrifuge tube containing 35 µl of

washed magnetic beads and immediately mixed by pipetting at least 10 times followed by vortexing the tube for 10 s at medium setting. The treated RNA sample and beads mix were incubated at room temperature for 5 min, after which the tubes were vortexed for 10 s at medium speed and incubated at 50°C for 5 min. Following this incubation, the tubes were immediately placed on a magnetic stand for at least 1 min and the supernatant was collected in a fresh 1.5 ml microcentrifuge tube. The rRNA-depleted samples were purified by adding 100 µl of RNase-free water, 18 µl of 3 M sodium acetate (Ambion), and 2 µl of 10 mg/ml glycogen, vortexed briefly followed by an addition of 600 µl of ice-cold isopropanol and incubated at –80°C overnight. On the following day, the precipitated RNA was pelleted and purified as described above, and the rRNA-depleted dephosphorylated footprint sample was dissolved in 10 µl of 10 mM Tris, pH 7.

## Library preparation from rRNA-depleted dephosphorylated footprint samples

The subsequent steps of linker ligation, reverse transcription, and circularization were carried out according to steps 30–46, and the PCR amplification and barcode addition steps were carried out as per steps 55–64 in the protocol by *Ingolia et al., 2012*. The resulting libraries were referred to as 'footprint libraries'.

## Library preparation for mRNA fraction

Total RNA was extracted from the tissue lysate reserved for mRNA sequencing using the hot-phenol method as described above. The extracted total RNA was submitted to Génome Québec (Montreal) for quality assessment and library preparation using the NEB mRNA stranded Library preparation service. The libraries thus generated are referred to as 'mRNA libraries'.

## Sequencing and analysis

Both the rFP and mRNA libraries were sequenced at an aimed sequencing depth of 50 million single-ended reads per sample on the Illumina HiSeq4000 or HiSeq2500 platforms. Demultiplexed sequencing data were provided by Génome Québec (Montreal) as .bam files.

## Bioinformatic analysis for ribosome footprinting data

The adapter sequence was identified using DREME-MEME Suite (*Bailey et al., 2015*) and trimmed from all the rFP reads using Trimmomatic (*Bolger et al., 2014*). Next, using Bowtie (*Bolger et al., 2014*), the rFP and mRNA reads were mapped to mouse reference genome mm10. Read counts for the uniquely mapped reads were generated and differentially translated genes were identified using the Xtail pipeline as described by *Xiao et al., 2016*. Transcriptionally and translationally regulated genes were identified by Xtail using default settings. Pathway analysis was carried out using Enrichr (*Chen et al., 2013*), using the Kyoto Encyclopedia of Genes and Genomes (*Kanehisa et al., 2017*) and Reactome (*Fabregat et al., 2017*) databases to identify cellular/molecular pathways associated with the differentially translated genes.

## Statistical analysis

GraphPad Prism v.9 software was used to analyze data. All data are presented as means ± SEM. An $\alpha$ = 0.05 was used to determine statistical significance. Data analysis included unpaired Student's $t$-test (two-tailed), one- and two-way ANOVA, and repeated measures ANOVA, followed by between-group comparisons using Tukey's post hoc test, as appropriate.

## Materials availability

This study did not generate new unique reagents.

## Acknowledgements

This study was supported by the Canadian Institutes of Health Research (PJT-870 162412) to AK, FRN-154281 to JSM, and NIH NINDS grant NS065926 to TJP. AK and CGG were supported by General Secretariat for Research and Innovation Greece ΤΙΕΡΑ 5-00024 (CGG), ERA-NET Neuron Sensory

disorders project TRANSMECH. KCL was supported by a Louise and Alan Edwards Foundation PhD fellowship.

## Additional information

### Competing interests

Hien T Zhao: Hien T. Zhao is a full-time employee and shareholder of Ionis Pharmaceuticals, Inc. The author has no other competing interests to declare. Bethany Fitzsimmons: Bethany Fitzsimmons is a full-time employee and shareholder of Ionis Pharmaceuticals, Inc. The author has no other competing interests to declare. The other authors declare that no competing interests exist.

### Funding

| Funder | Grant reference number | Author |
| --- | --- | --- |
| Canadian Institutes of Health Research | | Kevin C Lister<br>Calvin Wong<br>Sonali Uttam<br>Weihua Cai<br>David Ho-Tieng<br>Mehdi Hooshmandi<br>Ning Gu<br>Mehdi Amiri<br>Arkady Khoutorsky |
| Canadian Institutes of Health Research | PJT-870 162412 | Arkady Khoutorsky |
| Canadian Institutes of Health Research | FRN-154281 | Jeffrey S Mogil |
| NIH NINDS | NS065926 | Theodore J Price |
| General Secretariat for Research and Innovation Greece | T 12 E P A 5-00024 | Arkady Khoutorsky<br>Christos G Gkogkas |
| Louise and Alan Edwards Foundation PhD fellowship | | Kevin C Lister |

The funders had no role in study design, data collection, and interpretation, or the decision to submit the work for publication.

### Author contributions

Kevin C Lister, Conceptualization, Formal analysis, Investigation, Methodology, Writing – original draft, Writing – review and editing; Calvin Wong, Sonali Uttam, Investigation, Writing – review and editing; Marc Parisien, Formal analysis, Visualization, Writing – review and editing; Patricia Stecum, Nicole Brown, Weihua Cai, Mehdi Hooshmandi, Investigation; David Ho-Tieng, Mehdi Amiri, Seyed Mehdi Jafarnejad, Khadijah Mazhar, Formal analysis, Investigation; Ning Gu, Diana Tavares-Ferreira, Luda Diatchenko, Formal analysis, Investigation, Visualization, Writing – review and editing; Francis Beaudry, Formal analysis; Nikhil Nageshwar Inturi, Formal analysis, Visualization; Hien T Zhao, Bethany Fitzsimmons, Resources; Christos G Gkogkas, Nahum Sonenberg, Jeffrey S Mogil, Conceptualization, Writing – review and editing; Theodore J Price, Yaser Atlasi, Formal analysis, Writing – review and editing; Arkady Khoutorsky, Conceptualization, Resources, Supervision, Funding acquisition, Investigation, Visualization, Writing – original draft, Project administration, Writing – review and editing

### Author ORCIDs

Calvin Wong ⓘ https://orcid.org/0000-0001-7728-3035
Marc Parisien ⓘ https://orcid.org/0000-0003-2924-5960
Weihua Cai ⓘ https://orcid.org/0000-0003-2216-1422
Ning Gu ⓘ https://orcid.org/0000-0002-2433-0691
Seyed Mehdi Jafarnejad ⓘ https://orcid.org/0000-0002-5129-7081
Khadijah Mazhar ⓘ https://orcid.org/0000-0002-2469-7469

Nahum Sonenberg (iD) https://orcid.org/0000-0002-4707-8759
Theodore J Price (iD) https://orcid.org/0000-0002-6971-6221
Arkady Khoutorsky (iD) https://orcid.org/0000-0003-3846-8728

### Ethics

All procedures complied with the Canadian Council on Animal Care guidelines and were approved by McGill University's Downtown Animal Care Committee (protocol #7869).

Reviewer #2 (Public review): https://doi.org/10.7554/eLife.100451.3.sa1
Reviewer #4 (Public review): https://doi.org/10.7554/eLife.100451.3.sa2
Reviewer #5 (Public review): https://doi.org/10.7554/eLife.100451.3.sa3
Author response https://doi.org/10.7554/eLife.100451.3.sa4

## Additional files

### Supplementary files

Supplementary file 1. Ribo-seq and TRAP datasets.

Supplementary file 2. Sequences chemistry of ASOs.

MDAR checklist

### Data availability

Sequencing data generated in this study have been deposited in the Gene Expression Omnibus under the accession GSE265957.

The following dataset was generated:

| Author(s) | Year | Dataset title | Dataset URL | Database and Identifier |
|---|---|---|---|---|
| Lister KC, Wong C, Uttam S, Parisien M | 2024 | Translational control in the spinal cord regulates gene expression and pain hypersensitivity in the chronic phase of neuropathic pain | https://www.ncbi.nlm.nih.gov/geo/query/acc.cgi?acc=GSE265957 | NCBI Gene Expression Omnibus, GSE265957 |

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
