## [Editor Report · eLife Assessment]

Using a combination of innovative and robust techniques, this study outlines cell-type-specific translational landscape changes that occur in the spinal cord neurons in the early and late phases of nerve injury. The authors provided **compelling** evidence suggesting an essential role of protein synthesis regulation in the chronic phase of neuropathic pain. Although additional mechanisms contributing to late-phase neuropathic pain beyond altered PV+ neuron excitability remain to be elucidated, this is a **fundamental** and significant study toward a comprehensive understanding of the molecular pathways involved in neuropathic pain.

---

## [Referee Report · Reviewer #2 (Public review)]

Summary:

This manuscript compares transcription and translation in the spinal cord during the acute and chronic phases of neuropathic pain induced by surgical nerve injury. The authors chose to focus their investigation on translation in the chronic phase due to its greater impact on gene expression in the spinal cord compared to transcription.

(1) The study is significant because the molecular mechanisms underlying chronic pain remain elusive. The role of translational regulation in the spinal cord has not been investigated in neuroplasticity and chronic pain mouse models. The manuscript is innovative and technically robust. The authors employed several cutting-edge techniques such as Rio-seq, TRAP-seq, slice electrophysiology, and viral approaches. Despite the technical complexity, the manuscript is well-written. The authors demonstrated that inhibition of eIF4E alleviates pain hypersensitivity, that de novo protein synthesis is more pronounced in inhibitory interneurons, and that manipulating mTOR-eIF4E pathways alters mechanical sensitivity and neuroplasticity.

(2) Strengths: innovation (conceptual and technical levels), data support the conclusions.

Comments on revisions:

The authors did a great job addressing my comments.

---

## [Referee Report · Reviewer #4 (Public review)]

Summary:

The significance of this study lies in its focus on translational regulation in the late phase of neuropathic pain, using both genetic and pharmacological approaches, with specific emphasis on parvalbumin-positive (PV⁺) inhibitory interneurons in the spinal cord. The authors are very responsive to all the reviewers' comments.

Strengths:

I did not review this manuscript in the first round. However, the authors have been highly responsive to the reviewers' comments and have substantially strengthened the study. They conducted new behavioral experiments that yielded informative negative results (Fig. 6A and 6B). These findings demonstrate that targeting translational control in PV neurons is sufficient to reverse SNI-induced reductions in PV neuron excitability, but insufficient to ameliorate behavioral phenotypes. This suggests that additional cell types and pathways contribute to late-phase neuropathic pain.

Weaknesses:

Only the withdrawal threshold was measured to assess neuropathic pain. Some studies only used female mice. However, the authors appropriately discuss the study's limitations in the final two paragraphs and have added experimental details to improve clarity. Overall, the manuscript has been significantly improved.

---

## [Referee Report · Reviewer #5 (Public review)]

Summary:

This study investigates the molecular mechanisms underlying the maintenance of neuropathic pain, specifically focusing on the role of mRNA translation in the spinal cord. Using the Spared Nerve Injury (SNI) model, the authors demonstrate that while both transcription and translation are active in the early phase, the chronic phase (day 63) is uniquely characterized by a shift toward translational control. They identify spinal inhibitory neurons, particularly parvalbumin-positive interneurons, as key sites of this translational regulation.

Strengths:

Technical Rigor: The use of Ribo-seq and TRAP-seq allows for a high-resolution view of the "translatome," which more accurately reflects the functional protein output than standard mRNA-seq.Novelty: The study uncovers that reducing a single translation initiation factor (eIF4E) specifically in the CNS is sufficient to provide long-lasting relief from established chronic pain.Addressing Disinhibition: The electrophysiological evidence showing that increased translation in PV+ neurons reduces their excitability provides a clear mechanism for the "spinal disinhibition" typically seen in chronic pain.

Weaknesses:

Cell-Type Sufficiency: New experiments in the revision show that while inhibiting translation in PV+neurons restores their individual excitability, it is not sufficient on its own to reverse behavioral pain hypersensitivity. This suggests that the maintenance of chronic pain likely involves translational changes across a broader network of cell types, including other inhibitory neurons or non-neuronal cells like microglia. -This does not have to be resolved in the current study, but providing some framework to account for potential mechanisms might help the audience.

---

## [Author Response]

The following is the authors’ response to the original reviews

**Public Reviews:**

**Reviewer #1 (Public review):**
Summary:This study investigated the role of transcriptional and translational controls of gene expression in dorsal root ganglia and lumbar spinal cord in neuropathic pain in mice. Using ribosome profiling (Ribo-seq) and translating ribosome affinity purification (TRAP), they show changes in transcriptomic and translational gene expression at the peripheral and central levels rapidly after nerve injury. While translational changes in gene expression remained elevated for more than two months in both DRGs and the spinal cord, transcriptomic regulation was absent in the spinal cord long after the onset of neuropathy. Disrupting mRNA translation in dorsal horn neurons using antisense oligonucleotides reduced mechanical withdrawal threshold and facial expression of pain. Using fluorescent noncanonical amino acid tagging (FUNCAT), the authors further show that de novo protein expression primarily occurs in inhibitory neurons in the superficial dorsal horn after nerve injury. Accordingly, a selective increase in translational control of gene expression in spinal inhibitory neurons, or a subset of mainly inhibitory neurons expressing parvalbumin (PV), using transgenic mice, led to a decrease in the excitability of PV neurons and mechanical allodynia. In contrast, decreasing the translational control of spinal PV neurons prevented the alteration of the electrophysiological properties of the PV cells induced by nerve injury.Strengths:This is a well-written article that uncovers a previously unappreciated role of gene expression control in PV neurons, which seems to play an important part in the loss of inhibitory control of spinal circuits typically seen after peripheral nerve injury. The conclusions are generally well supported by the data.Weaknesses:The study would benefit from further clarifications in the methods section and a deeper analysis of gene expression changes in mRNA expression and ribosomal footprint observed after nerve injury.

We have improved the description of the methods and clarified the rationale underlying the presentation of gene expression changes. We have also added lists of the top differentially expressed genes at both the translational and transcriptional levels to Figure 1, and improved the description of the datasets in the Supplementary Materials.

Antisense oligonucleotides used to reduce translation by disrupting eIF4E expression were administered i.c.v. It is unknown if the authors controlled for locomotor deficits, which might add confounds in the interpretation of behavioral results. A more local route should have been preferable to avoid targeting brain regions, which could potentially affect behavior.

Thank you for raising this important point. We used i.c.v. administration to specifically target the central nervous system (CNS) without affecting the peripheral nervous system, as this is the recommended approach for selectively targeting the CNS using ASOs. Intraspinal administration of ASOs (into the spinal cord parenchyma) at an effective dose for long-term effects is not feasible. Intrathecal administration is possible but would result in exposure of the DRGs to the injected ASO and therefore would not be specific to the CNS.

To rule out potential locomotor deficits, we now subjected mice to the rotarod and open field tests to assess motor function. We found no differences between eIF4E-ASO– and control-ASO– injected mice (Fig. 2J, K).

In the revised version of the manuscript, we now better explain the rationale for i.c.v. injection. Moreover, we discuss the potential supraspinal effects of eIF4E-ASO in the Limitations section, while also describing the lack of motor phenotypes in the rotarod/open field tests.

Only female mice were used for Ribo-Seq, TRAP, FUNCAT, and electrophysiology, but both sexes were used for behavior experiments.

Our manuscript involves various complicated techniques and analyses. Due to limited resources, we therefore opted to use only females for expensive and labor-intensive experiments, such as Ribo-Seq, TRAP, FUNCAT, and electrophysiology, while using both sexes for behavioral studies.

We now clearly acknowledge this limitation in the revised manuscript.

The conditional KO of 4E-BP1 using transgenic animals should be total in the targeted cells. However, only a partial reduction is reported in Figure S2 in GAD2, PV, Vglut2, or Tac1 cells. Again, proper methods for quantification of fluorescence in these experiments are lacking.

We apologize for the oversight; we have now updated the description of the methods for IHC signal quantification. Although genetic ablation is indeed expected to result in a complete loss of signal, in practice, previous studies employing IHC, but not Western blotting, for 4E-BP1 have also shown only a partial reduction in signal. This is likely because the 4E-BP1 antibody partially detects other epitopes. Using the same antibody, we and others have shown complete elimination of the band corresponding to 4E-BP1 in spinal cord and DRG tissue (e.g., PMID: 26678009).

The elegant knockdown of eIF4E using AAV-mediated shRNAmir shows a recovery of the electrophysiological intrinsic properties of PV neurons after injury. It is unclear if such manipulation would be sufficient to reverse mechanical allodynia in vivo.

Thank you for this concern, which was also raised by other reviewers. We have now performed two additional experiments, which revealed that suppressing the mTORC1–eIF4E axis in spinal PV neurons (using AAVs expressing eIF4E-shRNA in spinal PV neurons [Fig. 6A] and transgenic mice expressing non-phosphorylatable 4E-BP1 in PV neurons [Fig. 6B]) is not sufficient to alleviate neuropathic pain. These new findings need to be reconciled with our other results showing that eIF4E downregulation in PV neurons prevents the SNI-induced reduction in their excitability, and that ASO-mediated suppression of eIF4E, which affects all cell types, alleviates neuropathic pain.

Together, these results suggest that targeting translational control in PV neurons is sufficient to reverse SNI-induced reduction in PV neuron excitability, but is not sufficient to prevent behavioral phenotypes, which likely require changes in other cell types and/or additional pathways, as well as other alterations within PV neurons. We have now included these new results in the revised manuscript (Fig. 6A and Fig. 6B) and revised the text accordingly. These changes include toning down the role of translational control in PV neurons after SNI in driving behavioral hypersensitivity.

**Reviewer #2 (Public review):**
Summary:I reviewed the manuscript titled "Translational Control in the Spinal Cord Regulates Gene Expression and Pain Hypersensitivity in the Chronic Phase of Neuropathic Pain." This manuscript compares transcription and translation in the spinal cord during the acute and chronic phases of neuropathic pain induced by surgical nerve injury. The authors chose to focus their investigation on translation in the chronic phase due to its greater impact on gene expression in the spinal cord compared to transcription.(1) The study is significant because the molecular mechanisms underlying chronic pain remain elusive. The role of translational regulation in the spinal cord has not been investigated in neuroplasticity and chronic pain mouse models. The manuscript is innovative and technically robust. The authors employed several cutting-edge techniques such as Rio-seq, TRAP-seq, slice electrophysiology, and viral approaches. Despite the technical complexity, the manuscript is wellwritten. The authors demonstrated that inhibition of eIF4E alleviates pain hypersensitivity, that de novo protein synthesis is more pronounced in inhibitory interneurons, and that manipulating mTOR-eIF4E pathways alters mechanical sensitivity and neuroplasticity.Strengths:Innovation (conceptual and technical levels), data support the conclusions.Weakness:Confusion about the sex of the animals. It is unclear whether eIF4E ASO affects translation and which cells. It is not determined that modulating translation in PV^+^ neurons impacts neuropathic pain behaviors.

We thank the reviewer for their thoughtful comments. In the revised version of the manuscript, we better explain that both sexes were used for behavioral experiments, whereas only females were used for Ribo-Seq, TRAP, FUNCAT, and electrophysiology experiments.

ASOs are not known to be intrinsically cell-type-specific; therefore, we do not expect differential effects on excitatory versus inhibitory neurons. We demonstrated that eIF4E-ASO reduces the levels of eIF4E, a key translation initiation factor that is rate-limiting for cap-dependent translation.

Moreover, in the revised manuscript we included two additional experiments (Fig. 6A and Fig. 6B) showing that decreased eIF4E-dependent translation in PV neurons is not sufficient to alleviate neuropathic pain, despite its effects on excitability measures. We have updated the manuscript to reflect these important new findings

**Reviewer #3 (Public review):**
Summary:This study provides evidence for translational changes in inhibitory spinal dorsal horn neurons following chronic nerve injury. Gene expression changes have been widely studied in the context of pain induction and provided key insights into the adaptation of the nervous system in the early phases of chronic pain. Whereas this is interesting biologically, most patients will arrive in the clinic beyond the acute phase of their injury, thus limiting the translational relevance of these studies. Recent studies have extended this work to highlight the difference between acute and chronic pain states, potentially explaining the cascading factors leading to chronic pain, and hopefully how to prevent this in vulnerable populations. The present study suggests that translational changes within spinal inhibitory populations could underlie long-term chronic pain, leading to decreased inhibition and heightened pain thresholds.Strengths:The approaches used and the broad outcomes of the manuscript are interesting and could be an exciting development in the field. The authors are using approaches more common in molecular biology and extending these into neuroscientific research, getting into the detail of how pathology could impact gene expression differentially across the course of an injury. This could open up new areas of research to selectively target not only defined populations but additionally help alleviate pain symptoms once an injury has already reached the maintenance phase. There is an opportunity to delve into what must be a very large data set and learn more about what genes are differentially translated and how this could affect circuit function.Weaknesses:Whereas the authors approach a key question in pain chronicity, the manuscript falls a little short of providing any conclusive data. The manuscript was in some areas very difficult to follow. Terminology was not always consistent or clear, and the flow of the manuscript could use some attention to highlight key areas. Whereas the overall message is clear in the summary, this would not necessarily be the case when reading the manuscript alone.

To improve the clarity and flow of the manuscript, we made changes to the text, including the addition of intermediate summaries and further explanations of terms and experiments.

The study claims to show that translational control mechanisms in the spinal cord play a role in mediating neuropathic pain hypersensitivity, but the studies presented do not fully support this statement. The authors instead provide some correlation between translation and behavioural reflex excitability (namely vfh and Hargreaves).It is difficult to fully interpret the work, as there are a number of inconsistencies, namely the range of timings pre- and post-injury, lack of controls for manipulations, the use of shmiRNA versus lineage deletions, and lack of detailed somatosensory testing. It is not completely clear how this work could be translatable as is, without a deeper understanding of how translational control affects circuit function and whether all of this is necessarily bad for the system, or whether this is a positive homeostatic adaptation to the hyperexcitability of the circuit following injury.A large portion of the work is focussed on showing an inhibitory-selective change in translation following chronic nerve injury. The evidence for this is however lacking. Statistics to show that translational effects are restricted to inhibitory subpopulations are inadequate. The author's choice of transgenic lines is not clear and seems to rely on availability rather than hypothesis.

Although we agree with some of the criticism, we have reservations regarding other points raised by the reviewer. To address several of the concerns, we added new experiments (Fig. 2J, 2K, 6A, and 6B). We also made changes to the text to improve readability and to better explain the rationale for the study and our focus on inhibitory neurons.

For example, we clarify that we do not state that changes in mRNA translation in the spinal cord during the chronic phase of neuropathic pain occur exclusively in inhibitory neurons. Although we observe changes in general protein synthesis, assessed using FUNCAT, in inhibitory but not excitatory neurons after SNI, alterations in the translation of specific transcripts, assessed using the TRAP approach, are observed in both excitatory and inhibitory neurons.

The second part of the paper focuses on inhibitory neurons because these neurons demonstrate larger translational changes. We now clearly indicate that alterations in excitatory neurons are also likely important during the chronic phase of SNI. This conclusion is further supported by newly added results (Fig. 6A and Fig. 6B), showing that targeting eIF4E-dependent translation in spinal PV neurons using two different approaches is not sufficient to reverse pain hypersensitivity.

**Recommendations for the authors:**

**Reviewer #1 (Recommendations for the authors):**
Analysis of gene expression in Figure 1 lacks clarity, and the data do not effectively guide the reader toward their intended purpose. A list of the most dysregulated genes at the transcriptional level, the translational level, or both, would help the reader fully appreciate the outcome of this analysis. Similarly, what is the message conveyed by Figures 4 D-G?

As requested, we have now included the top 10 upregulated and top 10 downregulated genes at both the translational and transcriptional levels in Figure 1. We also expanded the main text and figure legends to clarify that Supplementary Figure 1 includes volcano plots for all conditions, and that Supplementary Table 1 contains the complete datasets. In addition, we expanded the figure legends to explain the organization of the data in Supplementary Table 1. Finally, we provide pathway analyses of translationally regulated genes in the spinal cord, as this condition is the primary focus of the study.

Figure 4D–G shows the top 15 translationally upregulated and downregulated genes in inhibitory neurons at days 4 (D) and 60 (E), and in Tac1^+^ excitatory neurons at days 4 (F) and 60 (G) (four conditions in total) after SNI. These panels convey that translational regulation of specific transcripts occurs in both inhibitory and excitatory neurons. Panel 4H further demonstrates that, although translational changes are observed in both neuronal populations, a greater number of genes are altered in inhibitory neurons. We have improved the readability and flow of this section to better convey this message.

Details about how AHA was quantified in Figure 3 are missing. It is unclear how and where the cells were selected for quantification. Objective criteria for expression/no expression of AHA in the cells are not indicated. Additionally, the signal seems to have somehow been normalized over images from the contralateral side. It is difficult to understand what the bar graphs actually represent in panel C. One would interpret them as percentages of excitatory/inhibitory cells expressing AHA.

We apologize for the lack of clarity. We have now expanded the description of the analyses in the figure legend and in the Methods to better explain the results shown in Fig. 3. The imaged cells were selected based on specific criteria, such as lamina location and cell type. In panel C (the anisomycin experiment), values were normalized to the control group. In all other panels, no normalization was applied, and the values represent the AHA integrated density on maximumintensity projection images (averaged per mouse). We also describe the number of sections and cells per mouse, as well as other technical details, as requested.

In addition, a few minor changes should be made:(1) Rephrase Introduction: "Peripheral nerve injury can cause neuropathic pain, a chronic pain condition [...]." Neuropathic pain is not necessarily chronic.

This sentence was reworded to read “Peripheral nerve injury may result in neuropathic pain, a debilitating condition with limited effective treatment options”.

(2) Host species for secondary anti-mouse antibodies are provided but not for the anti-rabbit (donkey?). Also, check for consistency in the methods section. The method mentions P21 two secondary antibodies and an apparent third antibody named "anti-HRP-conjugated antibody." Please provide information about this antibody, or remove it.

Thank you for flagging it, the inadvertent repetition of “anti-HRP-conjugated antibody” was removed.

(3) Provide primary antibody hosts on page 22.

The hosts of all primary and secondary antibodies were now provided.

(4) Define PBST on page 21 and PBS-T on page 22.

We defined PBST in the revised manuscript (0.2% Triton-X100 in PBS).

(5) Specify the filter sets used for fluorescent microscopy.

We specified the filter sets used for fluorescent microscopy.

(6) Change the legend to 50% withdrawal threshold for vF behavior tests.

We addressed this by making the requested change in all relevant legends.

**Reviewer #2 (Recommendations for the authors):**
Major:(1) The authors need to show that eIF4E ASO (Figure 2) reduces translation in both inhibitory and excitatory neurons.

ASOs are not intrinsically cell-type specific, as they do not contain promoters or regulatory elements and act wherever they enter cells and engage RNase H1. However, differences in ASO effects across cell types can arise from variability in uptake, intracellular trafficking, RNase H activity, or target mRNA expression levels.

In our study, we used eIF4E-ASO as a general approach to demonstrate that eIF4E-dependent translation contributes to SNI-induced hypersensitivity, particularly at the chronic phase. We show a marked reduction in eIF4E levels in the spinal cord of eIF4E-ASO–injected mice compared with controls. We do not claim that the effects of eIF4E-ASO are mediated by a specific cell type; rather, they may involve excitatory neurons, inhibitory neurons, and non-neuronal cells, such as microglia and astrocytes, among others.

Notably, while eIF4E can promote general translation during development, in adult mice it predominantly regulates cap-dependent translation of specific mRNAs without having a major effect on overall protein synthesis. In our case, the partial reduction in eIF4E is unlikely to substantially affect general translation, as assessed by AHA incorporation, and would instead require TRAP or Ribo-Seq to detect transcript-specific translational changes. We now better explain the rationale for the eIF4E-ASO experiment and clearly state that the effects observed cannot be attributed to a specific cell type.

In addition, our new results showing that inhibition of eIF4E-dependent translation in PV neurons is not sufficient to alleviate SNI-induced mechanical hypersensitivity suggest that translational changes in other neuronal and/or non-neuronal cell types contribute to hypersensitivity. This important point is now more clearly explained in the revised manuscript, and the role of PV neurons is toned down throughout the paper.

(2) In Figure 5, it is necessary to show the effect of eIF4E-shRNA in PV+ neurons on neuropathic behaviors (von Frey and MGS).

To address this important concern, we performed two new experiments, both of which showed that inhibiting the mTORC1–eIF4E axis in parvalbumin neurons is not sufficient to alleviate neuropathic pain. First, we injected PV-Cre mice with AAV-eIF4E-shRNAmir and a scrambled control. We found that downregulating eIF4E in spinal PV neurons has no effect on SNI-induced mechanical hypersensitivity. We used a second, complementary approach to validate this finding. Specifically, we generated transgenic mice in which a non-phosphorylatable form of 4E-BP1 is expressed in PV neurons. Because non-phosphorylatable 4E-BP1 acts as a translational suppressor of eIF4E, this approach is functionally similar to eIF4E deletion.

Altogether, our findings indicate that cell-type–non-specific suppression of eIF4E using ASOs is sufficient to alleviate neuropathic pain, particularly at the chronic phase. In contrast, while activation of eIF4E-dependent translation in PV neurons (via 4E-BP1 deletion) induces pain hypersensitivity, suppression of eIF4E-dependent translation in PV neurons inhibits SNI-induced decrease in PV neuron excitability but does not alleviate pain hypersensitivity. Thus, increased eIF4E-dependent translation in PV neurons is sufficient to induce pain hypersensitivity, but targeting this pathway in PV neurons alone is not sufficient to reverse neuropathic pain.

Potential explanations for these findings include: (1) the presence of other important mechanisms in PV neurons (e.g., changes in synaptic transmission) that are translation independent; (2) the insufficiency of correcting reduced PV neuron excitability to alleviate hypersensitivity; and (3) an essential role for mRNA translation in other neuronal and/or non-neuronal cell types in neuropathic pain. We have updated the manuscript to include these potential explanations in the Discussion section.

Moderate:(1) In Figure 2, MGS should be performed at earlier time points as well.

We performed MGS when von Frey testing, which is less noisy and less labor intensive in our hands, suggested altered phenotypes.

(2) In Figure 4B, the gene markers are different in Gad2+ and Tac1+ cells. Please show the 12 markers for both cell types.

We now better explain the selection of the markers.

(3) In Figure 5, MGS should be performed to test if the effect is limited to mechanical sensation/reactivity or extends to nociception. Additionally, do these mice exhibit altered locomotion and grip strength?

As described above, we added experiments involving downregulation of eIF4E and expression of a mutant non-phosphorylatable 4E-BP1 in PV neurons. We performed von Frey testing, which showed no effect of suppressing the mTORC1–eIF4E axis on mechanical hypersensitivity under these conditions. Given these negative results, we did not proceed with mouse grimace scale (MGS) analysis.

(4) In Figure S2E, the reduction of eIF4E does not appear to be specific to GFP+ cells.

We now replaced the representative images in this Figure.

(5) Can chronic neuropathic pain be reduced by enhancing 4E-BP1 specifically in PV+ neurons?

We added the experiment proposed by the reviewer in Fig. 6B. We found that enhancing 4E-BP1 activity, by expressing a non-phosphorylatable form of 4E-BP1 in PV neurons, is not sufficient to alleviate neuropathic pain hypersensitivity.

(6) Why did the authors not use PainFace for the MGS?

We began using manual, blinded MGS scoring, as originally described by Mogil and colleagues in 2010 (PMID: 20453868), for this project before PainFace became available around 2019 (e.g., Tuttle and Zylka) and in later versions (e.g., PMID: 39024163). For consistency, we therefore continued using the same approach throughout the experiments.

(7) In Figures 2A-C, the labeling of the bar graphs seems incorrect: is it 4E-BP1 or eIF4E immunoreactivity?

Thank you very much for noticing this; we have corrected the mistake.

(8) In Figure 1, present the data by sex.

We performed sequencing analyses only in females. This decision was based on the large number of mice and experimental conditions required for both Ribo-Seq (n = 15 mice per replicate, 3 replicates per condition, and 2 time points for SNI/Sham, ~180 mice total) and TRAP (n = 3 mice per replicate, 3 replicates per condition, 2 time points, and 2 genotypes [Tac1 and GAD2] for SNI/Sham), as well as the high cost of sequencing. Behavioral experiments were performed in both sexes. This information is clearly indicated in the Methods section, and we have now also included it in the Limitations section of the paper.

(9) While the methods state that all behavioral testing was done with equal numbers of male and female mice, it seems that several experiments were done only in females. In the absence of a strong justification, all experiments should be conducted in both sexes.

As explained above, due to the very large number of mice required for some experiments and the high cost of sample processing and sequencing, only behavioral experiments were performed in both sexes. We now clearly describe the sex of the animals used in each experiment in the figure legends.

Minor:(1) In Figure 3, the legend is confusing and lacks labels.

We expanded the Fig. 3 legends and added labels, as requested.

**Reviewer #3 (Recommendations for the authors):**
Overall, the manuscript needs to be made clearer and more specific. As it stands, the logic and flow are difficult to follow. Figure legends are not always indicative of the figure and are inconsistent.Regarding timelines:The logic of the different timelines is not clear. Either explain why different times post-injury were chosen between experiments or keep them consistent. It seems a key message here is that the timing is important. It therefore follows that the authors should be strict about this in their own experiments. Figure 1: 4 and 63 days. Figure 2: Day 3 and weeks 8 and 12. Figure 3: Days 4 and 60. Figure 4: Days 4 and 60. Figure 5: 6 weeks. Figure S1: 4 and 60. Clarifying why these timings were used in each case and showing at the transcript level that these are most appropriate would be needed.

We thank the reviewer for carefully reviewing our manuscript. We focused on early versus late time points. For the sequencing experiments, we performed Ribo-seq at day 4 for the early time point and day 63 for the late time point, whereas TRAP analyses (and FUNCAT) were performed at day 4 for the early time point and day 60 for the late time point. These differences (day 60 versus day 63) were due to logistical issues related to sample collection. In our view, there are no major biological differences between day 60 and day 63 for the late time points, particularly because we do not perform direct comparisons across different experiments.

In other experiments, we used several time points (e.g., day 3, as well as 6, 8, and 12 weeks) either to follow the development of phenotypes or based on previous publications regarding the timing of specific effects. We now acknowledge the potential limitation of using slightly different time points in the Limitations section of the paper.

Regarding the use of inhibitory and excitatory markers:The comparisons they made between subpopulations seem a little random- for one, the number of Tac1 positive cells in the dorsal horn is not equal to that of PV, and so the comparison seems inappropriate.

The number of cells from each subpopulation should not affect the number of DEGs. Because these analyses were performed on bulk mRNA rather than at the single-cell level, the comparisons are made between SNI and control groups within each subpopulation. Thus, the number of differentially translated genes is determined per cell type, not per individual cell.

The lack of any semblance of variability or statistics with regard to gene changes makes it difficult to assess whether these comparisons were justified experimentally. Pax2 is a developmentally regulated transcription factor, with reduced levels in the adult. Using Pax2- NeuN+ to label excitatory interneurons is therefore not appropriate for comparison. A more appropriate comparison would be to use vGluT2 and GAD67. Similarly, the use of the GAD2Cre seems a poor choice. This is a restricted population of interneurons that have been suggested to have specific roles in presynaptic inhibition. If the authors were interested in this subpopulation for that reason, then they should state so.

Pax2 is commonly used as a marker of inhibitory neurons in the spinal cord (e.g. PMID: 36323322) as in the adult dorsal horn, Pax2 protein remains expressed in nearly all inhibitory neurons, including both GABAergic (GAD65/67^+^) and glycinergic (GlyT2^+^) neurons. VGluT2 marks terminals of IB4-binding peripheral sensory neurons as well as those of spinal cord excitatory interneurons in lamina II of the dorsal horn, complicating the analyses. We attempted using Lmx1b for excitatory neurons (Pax2 for inhibitory and Lmx1b for excitatory) but could not obtain specific and robust signal using different commercial antibodies (we have no access to non-commercial Pax2 antibody).

Regarding Cre lines, Gad2-Cre has been extensively used to target GABAergic neurons in the spinal cord. Although it is not expressed in purely glycinergic neurons, it is expressed in GABAergic and mixed GABA/glycine interneurons. Gad2-Cre is more restricted to superficial dorsal laminae I–III, which are relevant to pain processing, versus Gad1-Cre, which may also capture low-level GABAergic neurons in deep laminae and ventral horn inhibitory neurons. Moreover, there are also differences in the developmental profile, whereas Gad1-Cre is expressed earlier at embryonic stages during inhibitory neuron development, GAD2 is expressed later, in post-mitotic and mature inhibitory neurons. Because of these considerations (higher specificity to dorsal horn and later developmental expression), we used Gad2-Cre mouse line in our experiments.

Regarding cKO experiments:It is unclear whether the deletion of Eif4ebp (which is not "ablation" as stated in the manuscript) has had any effect on the PV/GAD2 cells themselves seeing as this deletion would be a lineage deletion. One would imagine that altering transcription in such a population from early development would affect a host of neuronal and circuit properties, such as connectivity, dendritic branching, etc. The authors should show that the circuit properties were not broadly changed, not least as PV is expressed throughout the nervous system and in muscles. This could in itself explain the hypersensitivity described in their results. Experimenters should repeat the AAV shRNAmir experiments in non-injured animals, and not just control animals with the scrambled sh.

We agree with the concerns related to potential developmental effects. Although it is nearly impossible to reliably and comprehensively demonstrate that circuit properties were not altered in our cKO mice, our manuscript presents several lines of evidence supporting a role for translational control in specific cell types in the regulation of gene expression and nociception independent of developmental effects. First, our translational gene expression analyses were performed in adult WT mice and reflect SNI-induced changes in gene expression at the translational level, assessed using complementary approaches. In addition, the effects of eIF4E ASO delivered to adult animals support a role for translational control in the regulation of SNI-induced pain hypersensitivity at later stages.

Moreover, downregulation of eIF4E in PV neurons using an AAV-based approach in adult mice affects their SNI-induced excitability, further supporting a role for translational mechanisms in regulating PV neuron plasticity after peripheral nerve injury in adulthood. To acknowledge the potential developmental effects associated with 4E-BP1 deletion using Tac1-Cre, Gad2-Cre, and PV-Cre mouse lines (with PV-Cre beginning expression postnatally), we have included an explicit limitation statement in the Discussion of the revised manuscript.

We also thank the reviewer for highlighting the distinction between deletion and ablation, and we have corrected this terminology in the revised manuscript.

Regarding pain:A large sticking point within the study is the lack of clarity of the populations they are targeting. Many of the populations mentioned are not expressed solely in the dorsal somatosensory horn and instead are also expressed in the ventral motor horn. This is particularly important with regard to the sensory tests they are performing, which rely on reflex responses. It seems these results, although interesting, are not proof of a pain effect, but rather showing changes in vfh-behaviour. To show this is a pain-specific event, and not just correlative or reflexive, the authors should perform further behavioural tests beyond vfh, Hargreaves, and the grimace scale, such as low threshold touch, rotarod, etc. How much of this effect is due to changes in reflex excitability? Would the authors expect similar results for all neuropathic models but not for chronic inflammatory states for example? Western Blot analysis at the moment is for the whole cord, which could imply changes in the ventral or intermediate horn, it could help strengthen the study to show that these changes are selective to the dorsal cord.

We have now added a new experiment showing that eIF4E-ASO has no effect on motor function in the rotarod and open field tests (Fig. 2J, K). In addition, the eIF4E-ASO experiment included in the original submission reflects supraspinal behavior, as assessed by MGS. Overall, our study includes numerous experiments and datasets. While we agree with some of the reviewer’s concerns, the extensive additional work requested, including additional neuropathic and inflammatory pain models, further assays of supraspinal behavior, Western blot analyses restricted to the dorsal horn, additional Cre lines and markers, and other analyses, is not feasible within the scope of the current manuscript.

Notably, in the revised manuscript, we have added new experiments (Fig. 2J, 2K, 6A, 6B) that we believe address the most critical concerns raised by the reviewers, and we have revised the text to more clearly acknowledge the limitations of the study.

Regarding patch clamp studies:An increase in rheobase alone in the PV cells would not in itself account for the changes seen in behaviour, seeing as the authors are suggesting this is a selective effect for von Frey and not radiant heat, for example. The authors should therefore show a change in mechanically-evoked firing of PV/GAD2 cells either by dorsal root stimulation in slice, or by cfos or equivalent marker of activation following sensory stimulation. The title of this figure is also misleading- it is not clear how there is any proof of promotion of plasticity in the experiments shown.

In the original submission, in addition to an increase in rheobase, we also demonstrated decreased spiking activity in response to a range of stimulating currents (Fig. 4). We agree that assessing mechanically evoked responses of PV neurons would be informative; however, such studies are beyond the scope of the current manuscript.

To address the final concern, we modified the title of Fig. 5 and the related text. Moreover, the newly added data showing that inhibition of translation in PV neurons does not alleviate SNIinduced hypersensitivity prompted us to tone down, throughout the manuscript, the link between translational changes in PV neurons and pain hypersensitivity.